# Revisiting Non-Parametric Matching Cost Volumes for Robust and Generalizable Stereo Matching

**Kelvin Cheng[†], Tianfu Wu[‡] and Christopher Healey[†]**
CS[†] and ECE[‡] at NC State University, Raleigh, NC 27695
{kbcheng, twu19, healey}@ncsu.edu

## Abstract

Stereo matching is a classic challenging problem in computer vision, which has recently witnessed remarkable progress by Deep Neural Networks (DNNs). This paradigm shift leads to two interesting and entangled questions that have not been addressed well. *First*, it is unclear whether stereo matching DNNs that are trained from scratch really learn to perform matching well. This paper studies this problem from the lens of white-box adversarial attacks. It presents a method of learning stereo-constrained photometrically-consistent attacks, which by design are weaker adversarial attacks, and yet can cause catastrophic performance drop for those DNNs. This observation suggests that they may not actually learn to perform matching well in the sense that they should otherwise achieve potentially even better after stereo-constrained perturbations are introduced. *Second*, stereo matching DNNs are typically trained under the simulation-to-real (Sim2Real) pipeline due to the data hungriness of DNNs. Thus, alleviating the impacts of the Sim2Real photometric gap in stereo matching DNNs becomes a pressing need. Towards joint adversarially robust and domain generalizable stereo matching, this paper proposes to learn *DNN-contextualized binary-pattern-driven non-parametric cost-volumes*. It leverages the perspective of learning the cost aggregation via DNNs, and presents a simple yet expressive design that is fully end-to-end trainable, without resorting to specific aggregation inductive biases. In experiments, the proposed method is tested in the SceneFlow dataset, the KITTI2015 dataset, and the Middlebury dataset. It significantly improves the adversarial robustness, while retaining accuracy performance comparable to state-of-the-art methods. It also shows a better Sim2Real generalizability. Our code and pretrained models are released at this Github Repo.

## 1  Introduction

Stereo matching remains a long-standing problem in computer vision that has been studied for several decades. As shown in the left of Fig. 1, given a pair of (rectified) stereo images, the reference left image $I^L$ and the right image $I^R$, for a pixel $I^L(x, y)$, its corresponding pixel in $I^R$ is constrained to be at $I^R(x - \mathcal{D}(x, y), y)$, where $\mathcal{D}(x, y)$ is called the **disparity** of the pixel $(x, y)$ in the reference left image. The disparity is inversely proportional to the depth $\mathcal{Z}(x, y)$, $\mathcal{D}(x, y) = \frac{f \cdot b}{\mathcal{Z}(x,y)}$, where $f$ is the focal length of the left camera and $b$ is the baseline (i.e., the distance between the two cameras). High-performing stereo matching will enable highly reliable and cost effective depth estimation from stereo images without leveraging expensive Lidar sensors, which has great potential in a wide range of applications such as autonomous driving and robot autonomy. To infer the disparity, the basic approach is to first compute the matching costs between $I^L(x, y)$ and $I^R(x - d, y)$ for different values of the disparity $d$ and then find the best match that corresponds to the minimum cost. Due to occlusions and textureless regions, the pixel correspondences can not be solved individually and the disparity map needs to be inferred in its entirety via prior-constrained (e.g., local disparity

36th Conference on Neural Information Processing Systems (NeurIPS 2022).

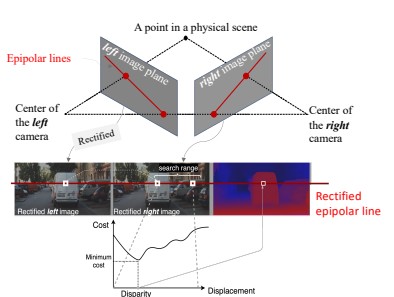 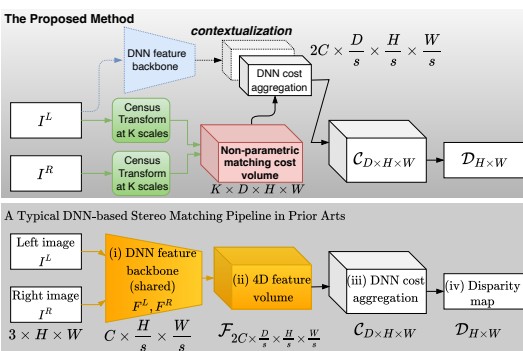

Figure 1: **_Left_**: Illustration of the stereo matching problem, which conventionally emphasizes the matching component between the (reference) left image, $I^L$ and the right image, $I^R$. **_Right-Bottom:_** A typical workflow of DNN-based stereo matching in the prior art. **_Right-Top:_** The proposed method casts stereo matching as a cost aggregation/optimization problem (solved by training a DNN) over a non-parametric cost volume (that truly focuses on matching) with (optional) parametric contextual features from the reference left image only. The non-parametric cost volume is realized by the census transform (CT) features [1]. CT enables blocking the gradients in learning white-box adversarial attacks (or significantly eliminating the effects even when a differentiable approximation of CT is used). Due to the binary nature of CT, it also helps improve the Sim2Real generalizability. To handle occluded and/or textureless regions better, the proposed method also utilize the contextual features computed from the left image only. See text for details.

smoothness) global matching cost optimization, which remains a challenging problem. Stereo matching has recently witnessed remarkable progress using Deep Neural Networks (DDNs). As shown in the right-bottom of Fig. 1, DNN-based stereo matching methods amortize the need of inducing inductive biases and of designing global matching cost optimization algorithms, which often consist of four components (see Appendix. A).

For the end-to-end fully differentiable learning of stereo matching in the prior art, two interesting questions arise: *First*, **how well do DNNs that are trained from scratch learn to match**? We can do some deductive reasoning: If they indeed learn to perform matching, their performance should potentially increase after we add the same perturbations at $I^L(x,y)$ and $I^R(x - \mathcal{D}(x,y), y)$ using the ground-truth disparity $\mathcal{D}(x,y)$ (i.e., the stereo-constrained photometric consistency), regardless of what the perturbations are (e.g., adversarial attacks). As we shall show, DNNs do not learn to match well and state-of-the-art stereo matching DNNs are vulnerable even when the photometric consistency is preserved against adversarial attacks (Section 2). *Second*, parallel to the adversarial vulnerability, **cross-domain generalizabilty** also is an important problem in stereo matching: DNN-based stereo matching is typically pre-trained under the so-called simulation to reality (Sim2Real) pipeline due to the high cost of collecting ground-truth matching results in practice and the data-hungry aspect of DNNs. It has been shown that DNNs may learn shortcut solutions that are strongly biased by the training dataset [2]. At the same time, it is desirable to have stereo matching systems that are more directly transferable from the simulation (training) to the reality (testing). Since stereo matching methods are widely used in autonomous driving, adversarial vulnerabilities in these models can lead to catastrophic consequences, and boosting the Sim2Real generalizability can significantly improve the applicability in diverse driving scenarios. Thus, jointly addressing adversarial vulnerabilities and the Sim2Real generalizability has become a pressing need in DNN-based stereo matching, as well as many other deep learning applications.

To address the two questions jointly in stereo matching, one key is to enable learning to really perform matching based on features that are robust and generalizable between the simulated data (training distributions) and real data (testing distributions). To that end, this paper proposes **DNN-contextualized binary-pattern-driven non-parametric cost-volumes**. On top of that, it revisits the perspective of learning the cost aggregation via DNNs for stereo matching, and presents a simple yet expressive design that is fully end-to-end trainable, without resorting to specific aggregation inductive biases. We briefly summarize the proposed methods as follows.

To defend against adversarial attacks, most methods rely on adversarial training [3], often at the expense of decreasing performance on clean images, long training time, and potential over-fitting to specific attacks and datasets (*e.g.* cannot transfer the robustness to other datasets as shown in the later sections). In contrast, **we propose to utilize domain-specific knowledge to facilitate the**

**built-in robustness of the neural networks.** Because of the strong *photometric consistency between stereo images*, stereo matching provides an ideal case to defend against adversarial attacks through the design of the neural network (see the right-top of Fig. 1, and Fig. 5 and Table 7 in the Appendix).

**For non-occluded regions** in stereo images, the corresponding pixels of the same physical point have similar colors. We observe that by using DNN features for matching, attacks will increase the matching costs for features that belong to the same physical point (see the appendix for details). Therefore, we propose to remove DNN features for matching and use hand-crafted features that will preserve the color differences to construct the cost volume, then the DNN is used to aggregate/optimize a non-parametric cost volume. To further make the cost as hard to alter as possible, we use local binary patterns that compare each pixel intensity to their neighbors (i.e., Census Transform [1, 4]) as the feature descriptor. **For occluded regions,** the photometric consistency does not hold. **For textureless non-occluded regions**, the local photometric features are weak for matching, even though the underlying consistency is retained. Thus, high-level semantic information learned by DNN features will be entailed. We use a DNN feature backbone for the reference (left) image only to contextualize the feature volume, which, albeit being fully differentiable, turns out not to hurt the adversarial robustness (see Sec. 4.2). This shows that the majority of the vulnerability actually comes from the matching part rather than the end-to-end trainable contextual information.

The non-parametric feature volume and the contextual DNN features will be fed through a head sub-network playing the role of a learnable optimizer that seeks the best matching result (Fig. 1). In essence, we cast stereo matching as a cost aggregation/optimization problem over a non-parametric cost volume with parametric contextual features. In experiments, we show that this more transparent approach improves adversarial robustness significantly while maintaining high accuracy.

In regards of boosting the Sim2Real generalizability, removing the DNN feature backbone for matching and utilizing the binary patterns of CT will induce the DNN to be a more general cost volume optimizer, thus alleviating the opportunity of shortcut learning in the feature space and resulting in better performance in cross-domain deployment, especially when no fine-tuning is used. These are verified in our experiments from the SceneFlow dataset [5] to the KITTI benchmark [6] and the MiddleBury dataset [7] when no fine-tuning is used.

With a comprehensive study of the proposed method (Section 4), **we have five observations as follows**: (1) By removing feature backbones (i.e., (ii) in the right-bottom of Fig. 1) that are trained on certain datasets and using non-parametric cost volumes as inputs, we can improve the cross-domain genearlizability through training the DNN as a general optimizer. (2) Even without contextual information (i.e., the component shown in dashed arrows and boxes in the right-top of Fig. 1), DNNs can do reasonably well on aggregating/optimizing the raw cost volume, which is nontrivial due to the loss of image information. (3) The vulnerability mostly comes from the matching part instead of the learned contextual information. (4) Since the vanilla CT can block the gradients in learning attacks, the obfuscated gradient problem [8] of CT is studied using a differentiable approximation. With this approximation, the overall DNN output can still be altered, showing the internal vulnerability of DNNs. (5) The robustness of adversarial training in stereo matching does not transfer well to other datasets or attacks, thus inducing the built-in robustness is very important, as done by the CT based non-parameteric cost volume in our proposed method (Section 3.2).

**Our Contributions.** This paper makes two main contributions to the field of stereo matching: (i) It proposes a novel design for stereo matching by utilizing DNNs to aggregate/optimize non-parametric cost volumes with parametric contextual features, which shows significantly better adversarial robustness and improved cross-domain (Sim2Real) generalizability when no-fine tuning is used. (ii) It presents the stereo-constrained projected gradient descent (PGD) attack method, which by design preserves photometric consistency to show the more serious vulnerabilities of state-of-the-art DNN-based stereo matching methods.

## 2 Attacking Stereo Matching Deep Neural Networks

### 2.1 The Proposed Stereo-Constrained PGD Attacks

To study the brittleness of DNN based stereo matching models, we intentionally develop a physically realizable attacking method based on the widely-used white-box PGD method [3], which retains the underlying photometric consistency in stereo matching by changing the pixel values of the same physical point in both images. Given a perturbation map $P(x, y), (x, y) \in \Lambda$ ($\Lambda$ denotes the image

| | EPE | | | | Bad 1.0 | | | | Bad 3.0 | | | |
|---|---|---|---|---|---|---|---|---|---|---|---|---|
| | CL | CT-0.03 | CT-0.06 | UCT-0.03 | CL | CT-0.03 | CT-0.06 | UCT-0.03 | CL | CT-0.03 | CT-0.06 | UCT-0.03 |
| PSMNet [9] | 0.28 | 29.05 | 84.04 | 91.08 | 2.00 | 84.75 | 90.41 | 92.75 | 0.16 | 54.80 | 83.68 | 89.91 |
| GANet-Deep [10] | **0.25** | 3.93 | 9.75 | 23.75 | **1.42** | 70.64 | 84.68 | 89.48 | **0.10** | 29.94 | 68.70 | 79.11 |
| LEAStereo [11] | 0.37 | 4.02 | 11.38 | 14.71 | 4.54 | 71.20 | 83.24 | 82.42 | 0.42 | 29.09 | 63.61 | 64.31 |
| Ours w/o ctx. | 0.38 | 1.13 | 1.43 | 2.36 | 4.14 | 24.64 | 30.69 | 41.34 | 0.32 | **2.46**(↑26.63) | 8.05 | 16.30 |
| Ours | **0.36** (↓0.11) | **0.88**(↑2.05) | **1.16**(↑8.59) | **1.81**(↑12.90) | **3.61**(↓2.19) | **21.20**(↑49.44) | **29.19**(↑54.05) | **36.42**(↑46) | **0.27**(↓0.17) | 3.75 | **6.17**(↑57.44) | **11.29**(↑53.02) |
| PSMNet + adv. | 0.46 | 0.70 | 1.02 | 1.06 | 8.04 | 17.78 | 33.54 | 36.50 | 0.66 | **1.40** | **3.08** | **4.14** |
| GANet + adv. | **0.42** | **0.65** | 0.98 | 1.05 | 6.47 | 14.99 | 28.56 | 31.22 | **0.63** | 1.40 | 3.76 | 4.36 |
| LEAStereo + adv. | 0.51 | 0.81 | 1.23 | 1.30 | 9.89 | 21.73 | 38.72 | 42.06 | 0.99 | 2.34 | 5.59 | 6.07 |
| Ours w/o ctx. + adv. | 0.42 | 0.78 | 0.90 | 1.26 | 5.95 | 16.83 | 21.42 | 32.27 | 0.73 | 2.88 | 3.83 | 7.51 |
| Ours + adv. | **0.41**(↑0.01) | **0.61**(↑0.04) | **0.69**(↑0.27) | **0.88**(↑0.17) | **5.77**(↓0.7) | **13.46**(↑1.53) | **16.29**(↑12.27) | **22.93**(↑8.29) | **0.52**(↑0.11) | **1.39**(↑0.01) | **2.00**(↑1.08) | **3.99**(↑0.15) |

Table 1: Stereo-constrained and unconstrained PGD attack results in the KITTI2015 training dataset [6]. For each metric (Section 4), the four columns show that metric on *CL*ean images, stereo-constrained attacked images (*CT*, $\epsilon = 0.03$, and *CT*, $\epsilon = 0.06$), and unconstrained attacked images (*UCT*, $\epsilon = 0.03$). **Row 1 & 3:** State-of-the-art stereo matching methods have catastrophic performance drop w.r.t. PGD attacks, and adversarially trained versions can not counter the drop. The results are tested in the KITTI2015 training dataset [6]. **Row 2 & 4** (see Sec. 4.2): Compared with the best result (in bold) of each metric by the prior art in Row 1 and 3 respectively, without adversarial training, the proposed method obtains slightly worse performance (as shown by ↓) on clean images, but achieves significantly better adversarial robustness (as shown by the ↑). With adversarial training, the proposed method consistently outperforms the prior art on both clean images and attacked images.

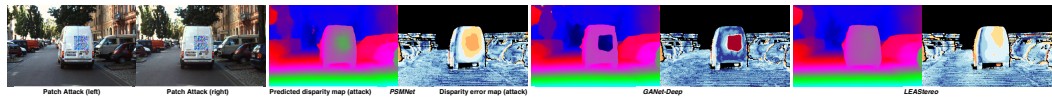

Figure 2: Illustration of the physical realizability of the proposed stereo-constrained attack method using adversarial patch attacks. *Best viewed in color and magnification.*

lattice), the distorted pixel values for each pixel location $(x, y)$ are computed as:

$$I_{adv}^L(x,y) = I^L(x,y) + P(x - \mathcal{D}(x,y), y), \quad I_{adv}^R(x,y) = I^R(x,y) + P(x,y), \qquad (1)$$

where $\mathcal{D}(x,y)$ is the ground-truth disparity map, and occluded areas will not be modified. Since the left image is the reference image for computing the disparity loss, we disallow to attack and evaluate occluded regions of the reference image. We note that by excluding occluded pixels, our goal is to show that the matching component is still vulnerable even with this constraint, and thus seeking more robust matching components is a pressing need in robust stereo matching. We also provide experiments on unconstrained attacks for comparisons.

Intuitively, this is a weaker attack method by design compared with the unconstrained counterpart, since it actually leaks the ground-truth disparity information to the algorithm using the pair $(P(x,y), P(x - \mathcal{D}(x,y), y))$ which may potentially help stereo matching, instead of attacking it. Consider two corresponding patches with constant disparity on the left and the right images containing the same physical points, the absolute sum of difference between these two patches will remain the same after the attack. The proposed stereo-constrained attacks uses the ground-truth disparity map in generating the perturbations, which is designed (i) to understand better the underlying vulnerability in stereo matching methods, and (ii) to simulate the physical scenarios, as the adversarial stereo-constrained patch attack experiments shown in Section 2.2.

We use the $L_\infty$ norm to measure similarities between images. Two images will appear visually identical under a certain threshold. To learn a $L_\infty$ bounded adversarial perturbation $P^{adv}$, the iterative PGD method is used,

$$P_{t+1}^{adv} = clip_P^\epsilon \{P_t^{adv} + \alpha \cdot sign(\nabla_P \ell(P_t^{adv}))\}, \qquad (2)$$

where $t = 0, 1, 2, \cdots, T$ and $P_0^{adv}$ starts with all zeros. $\ell(\cdot)$ denotes the mean absolute error for a perturbation $P_t^{adv}$ between the predicted disparity map for the perturbed images (Eqn. 1) and the ground-truth disparity map. And, $clip_P^\epsilon$ clips the perturbation to be within the $\epsilon$-ball of the corresponding zero-plane and the maximum color range. In our experiments, we set $\epsilon = 0.06$ or $0.03$ (the larger it is, the stronger the attack is), $\alpha = 0.01$ and $T = 20$. Appendix C.1 shows a toy example.

## 2.2 Attacking Results in KITTI2015

We test the proposed stereo-constrained PGD attack method and the unconstrained counterpart using three state-of-the-art methods in the KITTI2015 benchmark dataset. Table 1 (Row 1 & 3) shows the results. Since different methods use different training and validation splits when finetuning in the KITTI benchmark, the performance in Table 1 are computed on the entire dataset (200 images in total). Both the stereo-constrained attacks and the unconstrained ones can lead to catastrophic drop of performance. Such vulnerability may raise serious concerns for the deployment of DNNs in safety critical applications. The three methods cover DNN architectures by both sophisticated manual

design and differentiable neural architecture search. These strong attack performance clearly call for alternative designs of stereo matching networks that can exploit the nature of stereo matching for better robustness. We can see that the stereo-constrained attacks are indeed relatively weaker attacks.

**The Physical Realizability of the Proposed Stereo-Constrained Attack.** To test if the adversarial vulnerability can be intentionally exploited in a more realistic setting, such as autonomous driving, we constructed the patch attack experiment to demonstrate the possibility of such attempts. We select 10 scenarios where $40 \times 40$ adversarial patches can be put on more flat surfaces (Fig. 2, details and all images are provided in Appendix C.2). To preserve the depth of the scene, the ground truth disparities of the patches are the same as the corresponding part of the original image. For each image pair, we apply stereo-constrained PGD attacks with $\epsilon = 0.03$ and $T = 100$. Table 2 shows the results supporting the physical realizability.

## 2.3 Adversarial Training

To further support the pressing need of exploring alternative networks towards robust stereo matching, we show that the prior art reinforced by adversarial training still do not perform sufficiently well against the attacks. To compare with adversarial training, we fine-tune each method on the KITTI2015 training images perturbed by 3-step unconstrained PGD attacks for 20 epochs [1], denoted as

| Models | Clean | | | Synthetic Patch Attack | | |
|---|---|---|---|---|---|---|
| | EPE | Bad 1.0 | Bad 3.0 | EPE | Bad 1.0 | Bad 3.0 |
| PSMNet | 0.28 | 2.30 | 0.21 | **0.80** | 15.41 | **4.46** |
| GANet-Deep | **0.25** | **1.46** | **0.12** | 1.84 | **12.52** | 9.09 |
| LEAStereo | 0.41 | 5.60 | 0.69 | 1.34 | 20.32 | 6.18 |
| Ours w/o ctx. | 0.46 | 5.51 | 0.63 | **0.48**(↑0.86) | **6.42**(↓6.1) | **0.81**(↑5.37) |
| Ours | **0.40**(↓0.15) | **4.79**(↓3.33) | **0.46**(↓0.34) | 0.54 | 7.58 | 0.85 |
| PSMNet + adv. | 0.58 | 12.56 | 1.44 | **1.44** | 26.55 | **8.59** |
| GANet + adv. | **0.55** | **11.24** | **1.32** | 2.31 | **23.5** | 11.98 |
| LEAStereo + adv. | 0.71 | 15.87 | 2.24 | 1.96 | 30.88 | 11.72 |
| Ours w/o ctx. + adv. | **0.52**(↑0.03) | **8.95**(↑2.29) | **1.03**(↑0.29) | **0.54**(↑1.42) | **9.45**(↑14.05) | **1.18**(↑7.41) |
| Ours + adv. | 0.53 | 11.25 | 1.17 | 0.58 | 12.01 | 1.40 |

Table 2: Stereo-constraine adversarial patch attack results in the KITTI2015 training dataset. Note that the errors are computed using the whole image, while only a small portion of the image is attacked. Numbers in the bracelets with ↑ or ↓ are the difference between the performance of our method and the best results by the prior art (Row 1 & 3 respectively).

**+adv.** in tables. The bottom of Table 1 and the bottom of Table 2 show the results of attacks at the image level and the patch level respectively. While being relatively more robust after adversarial training, the performance drops significantly on clean images. The proposed robust stereo matching method will significantly improve this.

# 3 The Proposed Stereo Matching Network

Strong attacking results on state-of-the-art stereo matching methods and the lack of sufficient correction via adversarial training in Section 2 motivate us to explore alternative design of stereo matching networks that enjoy built-in robustness. In this section, we present the details of the proposed method.

## 3.1 Harnessing the Best of Classic Designs and DNNs

Fig. 1 illustrates the proposed workflow consisting of four main components:

**Extracting Features and Computing the Feature Volume Using Multi-Scale Census Transform.** Most current stereo matching methods use DNN-based features to form the 4D feature volume. In terms of matching, DNNs can increase the uniqueness of the feature for each pixel, but they also suffer from the inherent adversarial vulnerability. In contrast, traditional methods often use simple window-based similarity functions to initialize the costs, then rely on the optimization or cost aggregation stage to integrate all local cost information [12]. Following the same philosophy, we propose to use hand-crafted feature descriptors and similarity functions that are less sensitive to adversarial perturbations to initialize the costs, then rely on DNNs to integrate the local cost information. Specifically, we want the

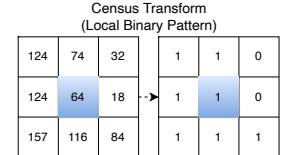

Figure 3: Illustration of the CT.

feature descriptor to change as little as possible when local intensities are perturbed. This specific requirement lead us to the Census Transform (CT), a traditional feature descriptor that is developed to eliminate the issue of radiometric differences caused by different exposure timing or non-Lambertian surfaces. Previous studies find that CT is the most robust and well-rounded cost function with global or semi-global methods [1, 4]. We use grey-scale raw intensity values in computing the census transform. Given a local window patch $R$ centered at a pixel $u \in \Lambda$ (Fig. 3), the census transform

---
[1]Due to the time-consuming adversarial training from scratch and the concerns of reproducing results of the prior art by retraining them from scratch, we use this post-hoc adversarial training strategy.

computes the local binary pattern by comparing each neighboring pixel $v \in R$ with $u$ such that it equals 1 if $I(v) >= I(u)$ and 0 otherwise. Hamming Distance (i.e. the number of different values in two bit strings) is used to compute the cost between two patches.

Unlike in traditional semi-global or global methods in which the cost of each pair can only be a scalar, we utilize multi-scale CTs to incorporate the context at different scales. We use local windows with sizes from $k_1$ to $k_2$ (e.g. $k_1 = 3, k_2 = 11$ in our experiments) so there are $K = k_2 - k_1 + 1$ costs associated with each matching candidate pairs, resulting in an initial 4D cost volume of the dimensions, $K \times D \times H \times W$. To normalize the cost at each scale, we divide the Hamming Distance by the number of pixels of each local window. To reduce the overall computational cost and the memory footprint, we use 3D convolutions to down-scale the cost volume to $1/3$ (i.e., $s = 3$ in Fig. 1) of its spatial size and the maximum disparity level, resulting the down-scaled cost volume of the dimensions, $C \times D_1 \times H_1 \times W_1$. Details of the architecture are included in the Appendix B.

**Boosting the Sim2Real generalizability.** Due to its binary nature, the cost curves will be less affected by the different color distributions of different datasets, thus improving the Sim2Real generalizability of the proposed stereo matching network, especially when no finetuning is used.

**Contextualizing the Feature Volume and Computing and Aggregating the Cost Volume.** Although being robust to adversarial attacks, the CT-based feature volume alone is not sufficiently powerful to handle occlusion, textureless (flat) regions, and more challenging semantic information, such as transparent objects and specular reflections. We introduce a 2-stack Hourglass [13] module with 2D convolutions to extract context information from the left reference image, resulting in a feature map of the same spatial size as the down-scaled CT cost volume, i.e., $C \times H_1 \times W_1$. The contextual feature map is unsqueezed and broadcasted along the second dimension to account for the down-scaled disparity levels, i.e., $C \times D_1 \times H_1 \times W_1$. The two are then concatenated along the first channel dimension (see Fig. 1). The contextualized cost volume will be fed into an encoder-decoder sub-network for the cost volume computation and aggregation stage, $\mathcal{C}_{D \times H \times W}$, which is realized by a 3-stack Hourglass [13] module with 3D convolutions. If the DNN feature backbone is not used to extract context information from the left image (Fig. 1), our model will be denoted as **"ours w/o ctx."** in tables and figures.

**Disparity Map Prediction.** To predict the final disparity map $\mathcal{D}(x, y), \forall (x, y) \in \Lambda$, the output of each module in the decoder (i.e., a stack in the Hourglass module) of the cost aggregation is first up-sampled to the original size $D \times H \times W$, denoted as $\mathcal{D}_m(d, x, y)$ where $m$ is the stack layer index. Then, similar to the method used in [14], the predicted disparity map $\mathcal{D}_m(x, y)$ is computed by,

$$\mathcal{D}_m(x, y) = \sum_{d=1}^{D} d \times Softmax(\mathcal{D}_m(d, x, y)), \tag{3}$$

where $Softmax$ is applied along the first disparity level dimension in $\mathcal{D}_m$.

**The Loss Function.** In training, we use the smooth $L_1$ loss, $\ell_{L_1}(z) = \frac{z^2}{2}, (\text{if } z < 1); |z| - 0.5, (\text{otherwise})$ due to its robustness at disparity discontinuities and low sensitivity to outliers [9, 10]. Given the ground-truth disparity map $\mathcal{D}^*(u)$, the loss is defined by,

$$\mathcal{L}(\Theta; \mathcal{D}^*) = \sum_{m=1}^{M} \beta_m \cdot \frac{1}{|\Lambda|} \sum_{(x,y) \in \Lambda} \ell_{L_1}(\mathcal{D}_m(x, y) - \mathcal{D}^*(x, y)), \tag{4}$$

where $\Theta$ collects all parameters in our model, $\beta_m$ represents the weight for the output from a stack layer $m$ (e.g., 0.5, 0.7, and 1 are used for the 3-stack Hourglass module in our experiments).

### 3.2 Attacking Census Transform and Its Built-in Robustness

Since Census Transform contains the non-differentiable comparison operator, the gradients from the constructed cost volume are blocked to the input images thus leading to an illusion of safety, *i.e.* the obfuscated gradient problem [8]. For fair comparisons with differentiable methods, we combine subtraction and the sigmoid function as a differentiable approximation of the comparison operator.

$$a > b \approx sigmoid(a - b) \cdot C \tag{5}$$

As suggested in [8], we should perform the usual forward pass through the comparison operator, but compute the gradients through the differentiable combination of subtraction and the sigmoid function on the backward pass. However, as the derivative of the sigmoid function is $(1 - x)x$, a boolean value (0 or 1) will make it zero, and thus blocking the gradient flow. Therefore, we use the sigmoid

| Models (trained on SceneFlow) | KITTI 2015 | | | KITTI 2012 | | | Middlebury | | |
|---|---|---|---|---|---|---|---|---|---|
| | EPE | Bad 1.0 | Bad 3.0 | EPE | Bad 1.0 | Bad 3.0 | EPE | Bad 1.0 | Bad 3.0 |
| PSMNet | 6.89 | 72.93 | 31.55 | 5.90 | 71.59 | 28.42 | 4.33 | 73.01 | 19.01 |
| GANet | 1.66 | 42.12 | 10.48 | 1.48 | 31.61 | 9.51 | 2.26 | 27.45 | 11.40 |
| LEAStereo | 2.00 | 51.29 | 13.90 | 1.91 | 44.26 | 14.28 | 3.47 | 32.67 | 14.81 |
| **Ours w/o ctx** | **1.25** | **25.95** | **6.12** | **1.23** | **19.66** | **6.80** | **1.71** | **18.72** | **9.16** |
| Ours | 1.26 | 27.92 | 6.31 | 1.28 | 20.62 | 7.16 | 1.96 | 20.09 | 10.05 |

Table 3: Comparisons for the Sim2Real cross-domain generalizability from the SceneFlow trained models to the KITTI 2015, KITTI 2012 and Middleburry datasets in testing without any fine-tuning.

in the forward pass as well and multiply the inputs by a large constant (e.g. $C = 10^5$) such that the output of the sigmoid function is close to either zero or one, while maintaining the gradient flow.

**Built-in Robustness of Census Transform.** Without using this differentiable approximation, our method without the contextual feature backbone will be ***unattackable*** since the gradient flows are completely blocked. From the perspective of attacks, the binary patterns generated by Census Transform is more difficult to alter due to the comparison operator. Given a threshold of maximum pixel difference in perturbation $\epsilon$, *neighbors will not be altered if their difference with the center is larger than* $2\epsilon$. Let $CT(u, v)$ denote the local binary value at $u$ with the neighbor $v$, $I$ be the image, and $CT'$ be the binary pattern after the perturbation, we have:

$$CT'(u, v) = CT(u, v), \quad \text{if } |I(u) - I(v)| > 2\epsilon \qquad (6)$$

If the attack does not violate photometric consistency, it will be even harder to alter the cost between binary patches of corresponding pairs. Specifically, if a neighboring pixel appears in both the left and the right binary patches, its relative magnitude relationship with the center pixel will be the same for both patches, no matter how its intensities change. It is our interest to test if this highly non-linear operator can defend the DNNs against attacks.

## 4 Experiments

We first present the results on the Sim2Real cross-domain generalizability, followed by showing results on the adversarial robustness. Implementation details are provided in the Appendix B.

*Data.* We evaluate our method on the SceneFlow [5] and KITTI2015 [6] datasets. We also test pretrained models on the KITTI2012 [15] and the Middlebury [7] dataset at quarter resolution. The SceneFlow dataset is a large-scale synthetic dataset that contains $35,454$ training images and $4,370$ test images at the resolution of $540 \times 960$. The KITTI2015 dataset is a real-world dataset of driving scenes, which contains 200 training images and 200 test images at the resolution of $375 \times 1242$. Since the depth of each scene is obtained through LiDAR, the ground truth is not dense.

*Evaluation Metrics.* We adopt the provided protocols in the two datasets. There are three metrics: **EPE [px]** which measures the end-point error in pixels, **Bad 1.0 [%]** and **Bad 3.0 [%]** which represents the error rate of errors larger than 1 pixel and 3 pixels respectively.

*Baseline Methods.* We compare with state-of-the-art deep stereo matching methods: the PSMNet [9], the GANet [10], and the LEAStereo [11]. We use their publicly released codes and trained model checkpoints in comparisons.

### 4.1 The Sim2Real Cross-Domain Generalizability

To verify the conjecture that cross-domain generalizability in stereo matching can be induced by removing the dependency between the cost volume computation and the dataset-dependent feature backbone, we evaluate all models pretrained on SceneFlow directly on the KITTI training datasets and the Middlebury training dataset [7]. As shown in Table 3, **our method outperforms prior art by a large margin**. This result shows that our proposed design of combining a non-parametric cost volume formed by the multi-scale census transform and a generalized cost aggregation/optimization DNN is indeed more cross-domain consistent. It also shows that the head sub-network DNN indeed learns to play the role of a domain-independent optimizer over a given cost volume.

### 4.2 The Adversarial Robustness

**Attacks at the image level.** To evaluate the adversarial robustness in KITTI2015, we directly test the trained models on the entire training dataset (200 images). Due to the GPU memory limitation, we only use the $240 \times 384$ center part of each image. Because of cropping, we also ignore those pixels where their correspondences are outside of the cropped images.

| Models | Clean | | | PGD Attack ($\epsilon = 0.03$) | | | PGD Attack ($\epsilon = 0.06$) | | |
|---|---|---|---|---|---|---|---|---|---|
| | EPE | Bad 1.0 | Bad 3.0 | EPE | Bad 1.0 | Bad 3.0 | EPE | Bad 1.0 | Bad 3.0 |
| PSMNet + adv. | 0.8 | 18.24 | 3.03 | 1.37 | 33.87 | 6.72 | 2.07 | 51.00 | 12.28 |
| GANet + adv. | 0.80 | 18.82 | 3.03 | 1.45 | 36.79 | 7.75 | 2.29 | 54.07 | 15.42 |
| LEAStereo + adv. | 0.85 | 20.17 | 3.56 | 1.48 | 38.24 | 8.60 | 2.3 | 55.67 | 16.24 |
| Ours w/o ctx. | 0.49 | 6.40 | 1.23 | 1.58 | 26.69 | 9.66 | 1.92 | 32.52 | 12.87 |
| Ours | 0.46 | 6.07 | 1.00 | 1.37 | 22.45 | 7.29 | 1.80 | 29.61 | 10.50 |
| Ours w/o ctx. + adv. | 0.65 | 11.18 | 2.25 | 1.28 | 24.62 | 6.64 | 1.48 | 28.93 | 8.31 |
| Ours + adv. | **0.56** | **10.14** | **1.73** | **0.89** | **18.35** | **4.14** | **1.09** | **22.64** | **5.95** |

Table 4: Transferrability of Adversarial Robustness: stereo-constrained 20-step PGD Attack Results in the KITTI2012 training dataset using adversarially trained neural networks on KITTI2015.

Table 1 (Row 2 & 4) shows the results. Compared with the prior art, **our method shows significantly better robustness on both stereo-constrained and unconstrained attacks.** We note that although the approximation used in attacking the CT is highly non-linear (Eqn. 5), adversarial attacks can still find ways to perturb the input images, which further demonstrate the vulnerability of the DNNs. As aforementioned, our method with the contextualized feature volume is more robust than its counterpart, showing that **the majority of the vulnerability actually comes from the matching part rather than the contextual information.** We note

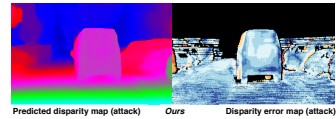

Figure 4: An example of adversarial patch attack on our method. Compared with Fig. 2, it is much more robust.

that although the context extractor is fully differentiable and thus vulnerable to attacks, to attack the CT-based matching component and the context extractor at the same time, an attaching method (e.g., the PGD) must find ways to alter both the cost volume and the context cues from the reference image. The fact that the robustness is not affected significantly shows that the alternation of the context cues either contradicts the alternation of the matching component, or being too weak compared to the matching component. In other words, the gradients from the context extractor have different directions or much smaller magnitudes compared to the gradients from the matching component. This finding shows a way to improve accuracy without sacrificing robustness in stereo matching. Those been said, the context extractor may still introduce the training domain dependence.

**Attacks at the patch level.** Table 2 shows the comparisons. Fig. 4 shows an example. *Our method is very robust against adversarial patches.* In contrast, other methods perform poorly, even with adversarial training.

### 4.3 Improvement via Adversarial Training

As shown in Table 1, our method without adversarial training shows comparable adversarial robustness with $\epsilon = 0.03, 0.06$, especially for EPE and Bad 1.0. In comparisons, for the prior art (Row 1 & 3), the difference between with and without adversarial training are significantly larger.

For the patch attack experiment, our method w/o adversarial training is already much more robust than the prior art with adversarial training (see Tabel 2). With adversarial training, our method has a stronger robustness than all other methods, showing that our approach is orthogonal to adversarial training and they could be jointly used to further improved robustness.

**Transferability of the reinforcement.** To show the importance of built-in robustness, we also test on KITTI2012 using models trained on KITTI2015 to see how the adversarial robustness generalize on unseen data. In Table. 4, our method shows a stronger cross-domain adversarial robustness than other adversarially trained methods. Similarly, our method with adversarial training is the most robust over all methods.

### 4.4 Ablation Study

**How important is the non-parametric cost volume?** To check whether the importance of the non-parametric cost volume can be justified, we modify our model by replacing the CT component with the concatenation of normal DNN features from the left and right images, while keeping other components unchanged (denoted by **"ours w/ feat."**). We compare with traditional Sum of Absolute Difference (SAD) and CT with a single scale. Table 5 shows that the ones with non-parametric cost volumes generalize much better than the one with DNN features. The contextualized CT feature volume is consistently better than the DNN feature volume.

**Why using the CT?** It is chosen due to its non-differentiability and the fact that it is a well-rounded choice in the literature. We use multi-scale representations to respect the common recognition of its expressivity, and to alleviate choosing window size as a dataset-dependent hyper-parameter. From Table 5, we can see that CT is indeed much more robust than SAD due to its non-differentiability. CT with multiple scales has a stronger robustness than the single-scale version, while having a slightly better accuracy due to its flexibility.

| Models | SceneFlow | | KITTI15 (Sim2Real) | | KITTI15 Attack ($\epsilon = 0.03$) | |
| --- | --- | --- | --- | --- | --- | --- |
| | EPE [px] | Bad 3.0 [%] | EPE [px] | Bad 3.0 | EPE [px] | Bad 3.0 [%] |
| multi-scale SAD | 1.02 | 4.02 | 1.71 | 9.69 | 2.30 | 18.20 |
| CT (w=11) | 1.18 | 4.77 | 1.28 | 6.38 | 1.88 | 7.22 |
| Ours w/ feat. | 0.85 | 3.75 | 3.25 | 26.76 | 2.13 | 15.23 |
| Ours w/o ctx. | 1.10 | 4.40 | **1.25** | **6.12** | 1.13 | **2.46** |
| Ours | **0.84** | **3.70** | 1.26 | 6.31 | **0.88** | 3.75 |

Table 5: Comparison between single-scale CT and multi-scale SAD.

| Models | Non-Occlusion | | All Areas | |
| --- | --- | --- | --- | --- |
| | FG | Avg All | FG | Avg All |
| GCNet [14] | 5.58 | 2.61 | 6.16 | 2.87 |
| PSMNet [9] | 4.31 | 2.14 | 4.62 | 2.32 |
| GANet-15 [10] | 3.39 | 1.84 | 3.91 | 2.03 |
| GANet-Deep [10] | **1.34** | 1.63 | **1.48** | 1.81 |
| LEAStereo [11] | 2.65 | **1.51** | 2.91 | **1.65** |
| Ours | 3.54 | 2.09 | 4.16 | 2.39 |

Table 6: KITTI2015 leaderboard using the Bad 3.0 [%] metric in the protocol.

### 4.5 KITTI Leaderboard Comparisons

Table 6 shows the comparisons. Our method is competitive compared to other state-of-the-art methods, while having much stronger adversarial robustness and cross-domain generalizability.

## 5 Related Work

**Deep Stereo Matching.** After the first deep learning approach for stereo matching was proposed in [16], the first end-to-end trainable DNN-based method (DispNet) was developed by [5], together with the synthetic SceneFlow dataset. The GCNet [14] further extends the end-to-end approach by concatenating features in the cost volume stage, using 3D convolutional layers for cost aggregation, and introducing the soft $\arg\min$ operator to compute the expected disparity. Most subsequent approaches followed these design choices and use the SceneFlow dataset in pretraining [6].

The cost aggregation stage was further studied in [9, 17, 18] using the Spatial Pyramid Pooling module for feature extraction and the stacked Hourglass structures [13]. In [19], fast stereo matching is studied by by building the cost volume purely using highly optimized hand-crafted features (e.g. Census Transform and Sum of Absolute Differences) at the expense of accuracy performance. They did not study adversarial attack and defense and the Sim2Real generalizability, which are the focuses of this paper. In [20], multiple hand-crafted features including CT are used for improving the Sim2Real generalizability, but did not study the contextualized settings as done in the proposed method. Spatial cost propagation layers are studied in [10, 21] to reduce the number of 3D convolutional layers. Neural Architecture Search techniques are used in [11] to automatically find optimal architectures for each stage and further improve the performance, which are the current state-of-the-art in the KITTI 2015 benchmark [6]. However, they significantly suffer from the proposed adversarial attacks.

The SGM-Nets [22] provide an elegant and well-designed integration between exploiting the SGM for more robust cost aggregation (handling occlusion) and leveraging DNNs for learnable hyper-parameter optimization (P1 and P2 in SGM). In terms of the integration strategy, our method shares similar spirits and motivations with SGM-Nets. The main differences lie in (i) the generic fully-trainable cost aggregation component in our method versus the SGM inductive bias, and (ii) with versus without contextualized cost volumes.

In [23], domain-invariant stereo matching networks were proposed using the "domain normalization" approach that regularizes the distribution of learned representations to allow them to be invariant to domain differences. In [24], the generalizability of stereo matching is addressed using multi-level cost volume and multi-scale feature constancy. In [25], cascade and fused cost volumes are used for robust stereo matching. While these works improve cross-domain generalizability, our approach is orthogonal to theirs (i.e., different normalization techniques and cost volume refinement strategies) and we believe they can be used together to further improve the cross-domain accuracy. From the perspective of adversarial attacks, the first two approaches focus on the cross-domain representation power of DNN features, which will potentially make them more vulnerable against attacks.

**Adversarial Attacks and Defense.** Assuming full access to DNNs pretrained with clean images, white-box targeted attacks are powerful ways of investigating the brittleness of DNNs. In autonomous driving, although physically realizable attacks are investigated in many tasks [26, 27, 28, 29, 30],

attacking stereo matching has not been well studied. [31] show that DNN-based stereo matching methods are vulnerable against unconstrained adversarial attacks on both images separately. Without enforcing photometric consistency, these attacks will violate the underlying physical properties of binocular vision and thus are not realizable in practice. For example, unconstrained attacks cannot compute adversarial patches to fool stereo systems. They also only focused on the relatively simpler FGSM attacks [32]. To find out whether stereo matching methods are indeed vulnerable in a physically realizable setting, we propose the stereo-constrained projected gradient descent (PGD) attack [3] and show that state-of-the-art methods are vulnerable even when the color differences between corresponding pixels are preserved. [27] studied adversarial patch based attacks in optical flow, which is inherently different to attacking stereo matching due to the underlyingly different matching nature of the problems. They did not study adversarial defense for optical flow methods. The neural matching paths [33] leverage neural paths in DNNs trained for classification for matching, which exploit low, middle and high level features jointly and thus are robust with respect to the style transfer experiments. Consider that DNNs trained for classification are also significantly vulnerable to adversarial attacks, we suspect that we could still learn to "fool" those DNNs in neural-path-based matching using the white-box PGD method.

Towards defense, adversarial training is the most widely used method to improve adversarial robustness [3, 34]. However, it also suffers from the disadvantages of dropping accuracy, long training time, and over-fitting to specific attacks and datasets. While adversarial training is universal to all kinds of DNNs, our method increases the built-in robustness by utilizing the photometric consistency of stereo matching, thus avoiding the mentioned disadvantages. It can also be combined with adversarial training to further improve robustness.

## 6 Limitations and Potential Negative Impacts of the Proposed Work

One main limitation is that while being more robust and Sim2Real generalizable, the performance of the proposed method on clean images has some room to improve. One direction is to leverage neural architecture search to find more suitable DNN aggregation component. One potential negative impact of the proposed attack method is that since it is easy to implement the proposed physically realizable attacks they could be used in some unintended way to computer vision systems relying on the conventional DNN-based stereo matching. Another limitation is that handcrafted CT features used in the proposed method may cause accuracy performance issues in multi-view stereo matching, where there can be strong viewpoint and illumination changes as the images can be taken at different points in time and by different cameras. It is not clear whether the proposed contextualized non-parametric cost volume and the generic Hourglass stack based cost aggregation could potentially address them, which we leave for future work.

## 7 Conclusions

This paper presents a novel design for stereo matching, which utilizes DNNs to aggregate/optimize non-parametric cost volumes with parametric contextual features. It harnesses the best of classic features (multi-scale census transform) and end-to-end trainable DNNs for adversarially-robust and cross-domain generalizable stereo matching. The proposed method is motivated by the observation that DNN-based stereo matching methods can be deceived by a type of physically realizable attacks that entail stereo constraints in learning the perturbation. In experiments, the proposed method is tested in SceneFlow and KITTI2015 datasets with significantly better adversarial robustness and Sim2Real cross-domain generalizability achieved.

## Acknowledgment

K. Cheng and T. Wu were supported by NSF IIS-1909644, ARO Grant W911NF1810295, NSF CMMI-2024688, NSF IUSE-2013451 and DHHS-ACL Grant 90IFDV0017-01-00. T. Wu was also supported by a research gift fund from the Innopeak Technology, Inc. (an affiliate of OPPO). We thank the reviewers and area chairs for their constructive comments. T. Wu also thanks Drs. Zhebin Zhang and Hongyu Sun at the OPPO research, Seattle for their helpful discussions improving the writing. The views presented in this paper are those of the authors and should not be interpreted as representing any funding agencies.

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
