## A    Background on Stereo Matching

The left of Fig. 1 illustrates the stereo matching setup. The rectified left image is used as the reference image to infer the disparity map. For each pixel in the left image, the goal of stereo matching is to find the target pixel on the rectified epipolar line in the right image. The search range (i.e., disparity levels) often is predefined and fixed to a sufficiently large value in the cost volume computation stage. The matching is based on minimizing the cost between features centered at the source pixel and the target pixel respectively. The challenge of stereo matching is to seek the globally optimal matching for all pixels in the left reference image and to handle many uncertainties such as in the appearance features (textureless regions and specularities), the cost function, and unknown repeated patterns and occlusion situations.

Let $\Lambda$ be an image lattice (e.g., $540 \times 960$) on which the rectified left and right images are defined, denoted by $I^L$ and $I^R$ respectively. Denote by $D(x, y)$ be the disparity map for the reference image $I^L$. In traditional methods, stereo matching is formulated as an energy/cost minimization problem,

$$\min_D E_d(I^L, I^R, D) + \lambda \cdot E_s(D), \tag{7}$$

where the first term is the data energy/cost, $E_d(I^L, I^R, D) = \sum_{(x,y)\in\Lambda} Cost(F_{I^L}(x, y), F_{I^R}(x - D(x, y), y))$ capturing the matching cost between a source pixel $(x, y)$ in the left reference image and the target pixel $(x - D(x, y), y)$ on the rectified epipolar line (i.e., the same row) in the right image. The cost is measured based on features $F_{I^L}$ and $F_{I^R}$ extracted for the source and target pixel respectively. The second term represents the prior/regularity of a disparity map such as the pairwise smoothness assumption, $E_s(D) = \sum_{(u,v)\in\mathcal{N}} S(D(u), D(v))$ where $u, v \in \Lambda$ and $\mathcal{N}$ the set of neighboring pixels (e.g., the 4-connected neighborhood). The challenges in the traditional formulation are in two-fold: what are the good features and the cost functions in the data term? And, what is the good prior that are sufficiently expressive to capture the disparity structures while facilitating efficient optimization (e.g., by the dynamic programming algorithm or semi-global method [35])?

Deep learning approaches mitigate the aforementioned challenges by exploiting the highly-expressive representational power and the end-to-end learning capability of DNNs. As shown in the right-bottom of Fig. 1, DNN-based stereo matching methods amortize the need of inducing proper priors (inductive biases) and of designing global matching cost optimization algorithms, which often consist of four components: (i) Extracting DNN features, $F^L$ and $F^R$ for matching with the spatial downsampling rate, $s$, (ii) Computing the 4D feature volume $\mathcal{F}$ by concatenating features $F^L(u, v)$ and $F^R(u - d, v)$ w.r.t. each disparity level $d \in [0, \frac{D}{s}]$ where $D$ is the predefined maximum disparity level, (iii) Computing and aggregating the matching cost volume $\mathcal{C}$ for each disparity level at the input resolution, which is typically realized via 3D convolution under an U-Net type of encoder-decoder architecture, which represents the solution space with respect to the data term in Eqn. 7, and (iv) Estimating the final disparity map $\mathcal{D}$. The prior/regularity term of a disparity map is made implicitly by the supervised loss function. The optimization algorithms (e.g., the traditional global or semi-global methods [35]) are also implicitly realized by a head sub-network (e.g., the stacked Hourglass sub-network in Fig. 5).

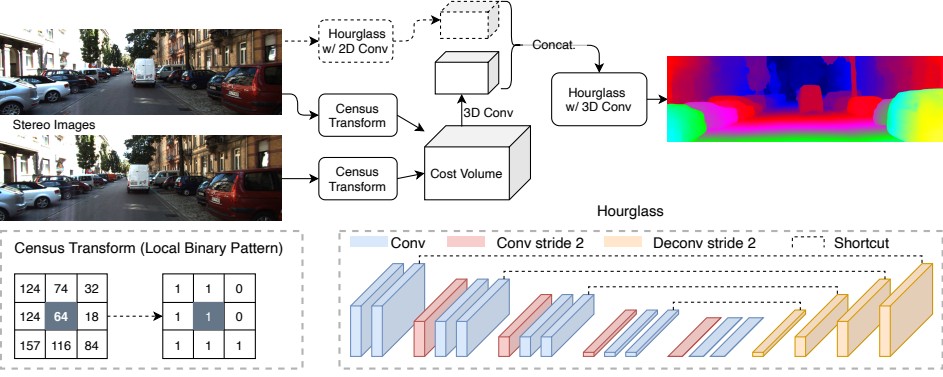

Figure 5: Network architecture.

# B   Network Architecture and Training Details

We show the detailed configuration of the proposed architecture in Table 7 with the workflow reproduced in Fig. 5. Besides how layers are wired, one key difference is the down-sample scale of the cost volume. While our method uses the same $1/3$ scale as GANet, PSMNet uses $1/4$ and LEAStereo uses $1/6$. Intuitively, a smaller down-sample size should lead to a stronger robustness, because more perturbations are averaged out in the cost volume. Therefore, our method is not taking advantage of the dowm-sampled size. In fact, it is much more robust than PSMNet and LEAStereo with a finer resolution.

*Implementation Details.* Our method is implemented in PyTorch and trained end-to-end using the Adam optimizer with $\beta_1 = 0.9$ and $\beta_2 = 0.999$. All images are preprocessed with color normalization. During training, we use a batch size of 8 on four GPUs (Tesla V100) using $240 \times 576$ random crops. The maximum disparity level is set to 192 and any values larger than this threshold will be ignored during training. For SceneFlow, we train our model from random initialization for 20 epochs with a constant learning rate of 0.001. For KITTI2015, we split the 200 training images into a training set of 140 images and a validation set of 60 images. We fine-tune our model pretrained on the SceneFlow with 600 epochs and use the validation set to select the best model.

| Name | Layer description | Output dimension |
|---|---|---|
| input | normalized image pairs | $H \times W \times 3$ |
| **Backbone** for the left reference image | | |
| conv_start | $3 \times 3$ Conv, stride 1
$5 \times 5$ Conv, stride 3
$3 \times 3$ Conv, stride 1 | $1/3H \times 1/3W \times 32$ |
| conv_1a | $3 \times 3$ Conv, stride 2 | $1/6H \times 1/6W \times 48$ |
| conv_2a | $3 \times 3$ Conv, stride 2 | $1/12H \times 1/12W \times 64$ |
| conv_3a | $3 \times 3$ Conv, stride 2 | $1/24H \times 1/24W \times 64$ |
| deconv_1a | $3 \times 3$ Deconv, stride 2,     add conv_2a | $1/12H \times 1/12W \times 64$ |
| deconv_2a | $3 \times 3$ Deconv, stride 2,     add conv_1a | $1/6H \times 1/6W \times 48$ |
| deconv_3a | $3 \times 3$ Deconv, stride 2,     add conv_start | $1/3H \times 1/3W \times 32$ |
| conv_1b | $3 \times 3$ Conv, stride 2,     add deconv_2a | $1/6H \times 1/6W \times 48$ |
| conv_2b | $3 \times 3$ Conv, stride 2,     add deconv_1a | $1/12H \times 1/12W \times 64$ |
| conv_3b | $3 \times 3$ Conv, stride 2,     add conv_3a | $1/24H \times 1/24W \times 64$ |
| deconv_1b | $3 \times 3$ Deconv, stride 2,     add conv_2b | $1/12H \times 1/12W \times 64$ |
| deconv_2b | $3 \times 3$ Deconv, stride 2,     add conv_1b | $1/6H \times 1/6W \times 48$ |
| deconv_3b | $3 \times 3$ Deconv, stride 2,     add deconv_3a | $1/3H \times 1/3W \times 32$ |
| backbone output | repeat deconv_3b $\ell/3$ times (dim=2) | $1/3H \times 1/3W \times 1/3\ell \times 32$ |
| **Multi-Scale Census Transform** | | |
| census_transform | census transform of the input with window size 11 | $H \times W \times 120$ |
| **Cost Volume** | | |
| init_cost_volume | the initial cost volume | $H \times W \times \ell \times 9$ |
| conv_3d_0a | $5 \times 5$ Conv3D, stride 3 | $1/3H \times 1/3W \times 1/3\ell \times 32$ |
| concat | [optional] concatenate with the backbone output (as context) | $1/3H \times 1/3W \times 1/3\ell \times 64$ |
| conv_3d_0b | $3 \times 3$ Conv3D | $1/3H \times 1/3W \times 1/3\ell \times 32$ |
| **Cost Aggregation** | | |
| conv_3d_1 | $[3 \times 3$ Conv3D, stride $1]_{\times 2}$ | $1/3H \times 1/3W \times 1/3\ell \times 32$ |
| conv_3d_2 | $3 \times 3$ Conv3D, stride 2
$[3 \times 3$ Conv3D, stride $1]_{\times 2}$ | $1/6H \times 1/6W \times 1/6\ell \times 32$ |
| conv_3d_3 | repeat above | $1/12H \times 1/12W \times 1/12\ell \times 32$ |
| conv_3d_4 | repeat above | $1/24H \times 1/24W \times 1/24\ell \times 32$ |
| conv_3d_5 | repeat above | $1/48H \times 1/48W \times 1/48\ell \times 32$ |
| deconv_3d_1 | $3 \times 3$ Deconv, stride 2,     add conv_3d_4 | $1/24H \times 1/24W \times 1/24\ell \times 32$ |
| deconv_3d_2 | $3 \times 3$ Deconv, stride 2,     add conv_3d_3 | $1/12H \times 1/12W \times 1/12\ell \times 32$ |
| deconv_3d_3 | $3 \times 3$ Deconv, stride 2,     add conv_3d_2 | $1/6H \times 1/6W \times 1/6\ell \times 32$ |
| cost_agg_1 | $3 \times 3$ Deconv, stride 2,     add conv_3d_1 | $1/3H \times 1/3W \times 1/3\ell \times 32$ |
| cost_agg_2 | repeat cost aggregation (input: cost_agg_1) | $1/3H \times 1/3W \times 1/3\ell \times 32$ |
| cost_agg_3 | repeat cost aggregation (input: cost_agg_2) | $1/3H \times 1/3W \times 1/3\ell \times 32$ |
| **Disparity Regression** | | |
| output_1 | $5 \times 5$ Deconv, stride 3 (input: cost_agg_1)
disparity regression (Eqn. 1 in the submission) | $H \times W \times 1$ |
| output_2 | repeat above (input: cost_agg_2) | $H \times W \times 1$ |
| output_3 | repeat above (input: cost_agg_3) | $H \times W \times 1$ |

Table 7: Details of the proposed network architecture. All convolution (Conv and Conv3D) and deconvolution (Deconv) layers are followed by batch normalization and ReLU.

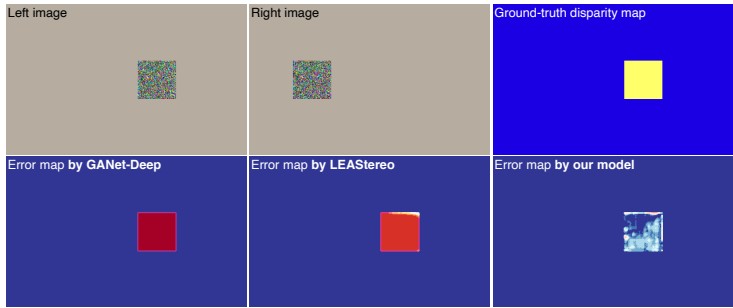

Figure 6: Illustration of adversarial vulnerability of deep stereo matching methods using a toy example: GANet-Deep [10], LEAStereo [11], and the proposed method. See text for detail.

| Models | Clean / After Attack, (EPE [px]) | | | | |
|---|---|---|---|---|---|
| | $\ell = 20$ | 60 | 100 | 140 | 180 |
| PSMNet | 1.89/78.85 | 1.22/63.57 | 0.45/3.29 | 0.94/4.59 | 0.48/36.10 |
| GANet | 2.36/9.94 | 6.74/27.72 | 7.37/80.39 | 13.56/115.77 | 15.09/31.95 |
| LEAStereo | 0.54/31.86 | 0.32/1.13 | 0.36/1.18 | 0.33/98.23 | 0.96/146.68 |
| Ours w/o ctx. | **0.057/0.051** | **0.20/0.20** | 0.29/**0.25** | **0.25/0.24** | 0.26/**0.26** |
| Ours | 0.36/3.12 | 0.28/5.56 | **0.13**/0.44 | 0.37/9.47 | **0.094**/1.77 |

Table 8: Result comparisons on synthetic adversarial patches at different disparity levels $\ell$.

## C  More Results

### C.1  A Toy Experiment

We conduct a toy experiment to show that state-of-the-art stereo matching methods can be easily attacked even by the simplest form of stereo-constrained attack, *i.e.* shifted patches (Fig. 6).

We create five synthetic toy stereo image pairs. In a stereo pair, the left reference image is composed by superposing a white-noise patch onto a constant background. The right image is created using the same patch and the same background in which the patch is horizontally shifted with respect to a given disparity level (such as $\ell = 20$). So, the ground-truth disparity for the entire patch will be the specified $\ell$. The background is excluded from the evaluation.

As shown in Table 8, for the clean synthetic images, state-of-the-art stereo matching methods work very well using the SceneFlow trained model checkpoints. Our model shows better performance for all disparity levels. After applying the proposed stereo-constrained PGD attack only to the

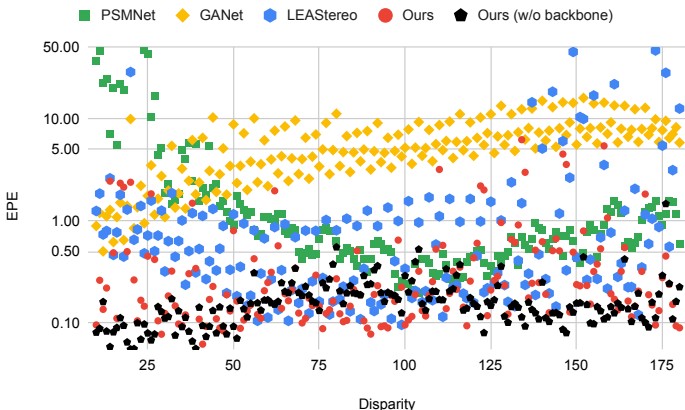

Figure 7: Test results of of shifting an adversarial patch on the left image at disparities from 10 to 180, while fixing the right image. Each point represents a testing pair with a different displacement.

patches (similar in spirit to the adversarial patches [36]), state-of-the-art methods' performance drop significantly except for the LEAStereo method [11] at two disparity levels (60 and 100). Fig. 7 further shows the effects of applying the same adversarial patch at different disparity levels, showing that this simplest form of attack has a certain transferability with different depth levels.

**Through this toy experiment, we can observe:** (i) Although not trained with this synthetic setting, state-of-the-art stereo matching methods are capable of recovering matching results when no attacks are applied. (ii) However, the matching capabilities are not stable even with respect to the much weaker stereo-constrained attacks. This may indicate that state-of-the-art methods could learn shortcut solutions in computing the cost volume, while our methods that directly utilize local rank information in computing the cost volume are more robust, either with or without the feature backbone in computing the cost volume.

## C.2 Examples of Adversarial Patch Attacks

We show all of the 10 scenarios for adversarial patch attack in figures 8, 9, 10, 11, and 12. All scenarios are selected where the adversarial patches can be put on more flat surfaces, but they are not necessarily horizontal to the image plane. Note that the ground truth disparities of the patches are the same as the corresponding part of the original image. The first row shows the attacked image pairs for our method. Other methods will have patches on the same location but the texture will be different. PSMNet [9], LEAStereo [11], and GANet [10], and our method are shown on the second, third, fourth, and fifth row respectively. The first and the third columns are the after attack disparity maps, and the second and the fourth columns are the after attack error maps from the ground truth.

From the results, we can see that our method is significantly more robust than others in this physically realizable attack setting.

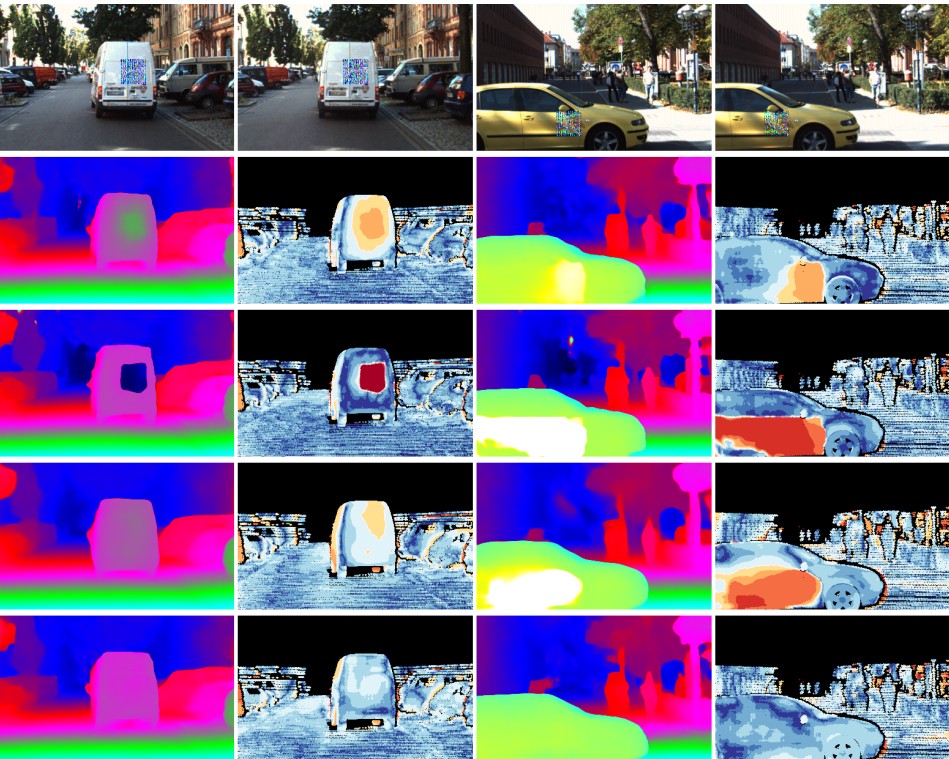

Figure 8: Illustration of the adversarial patch attack (1/5). The first row shows the attacked image pairs. PSMNet [9], GANet [10], LEAStereo [11], and our method are shown on the second, third, fourth, and fifth row respectively. The first and the third columns are the after attack disparity maps, and the second and the fourth columns are the after attack error maps from the ground truth.

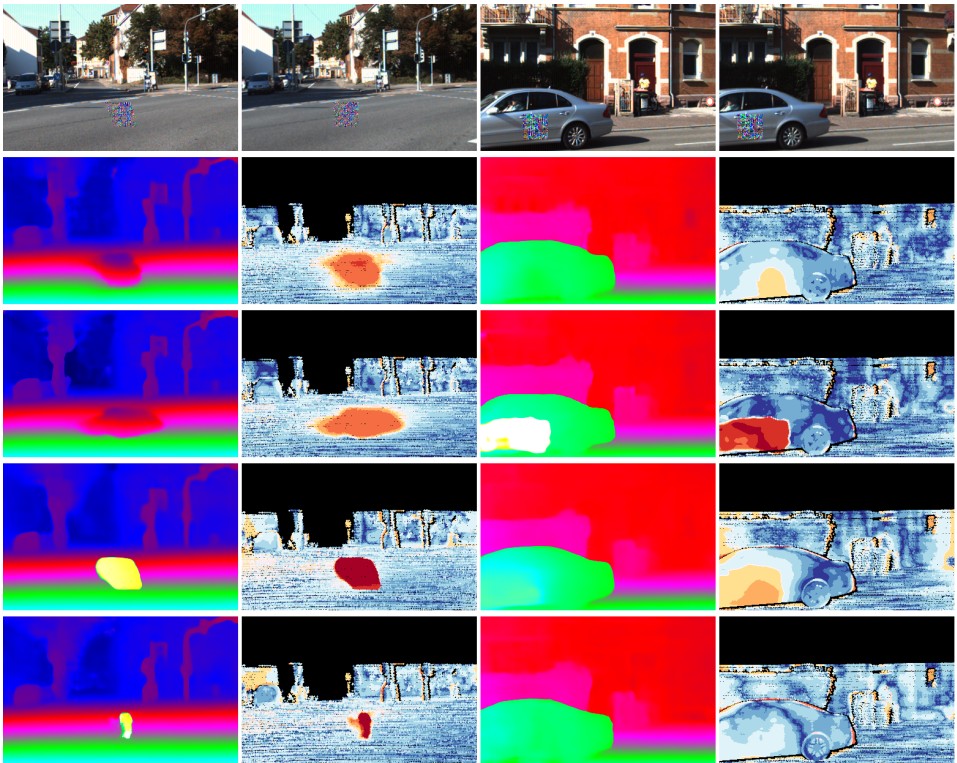

Figure 9: Illustration of the adversarial patch attack (2/5). The first row shows the attacked image pairs. PSMNet [9], GANet [10], LEAStereo [11], and our method are shown on the second, third, fourth, and fifth row respectively. The first and the third columns are the after attack disparity maps, and the second and the fourth columns are the after attack error maps from the ground truth.

| | clean | | | PGD Attack ($\epsilon = 0.03$) | | |
|---|---|---|---|---|---|---|
| Models | EPE | Bad 1.0 | Bad 3.0 | EPE | Bad 1.0 | Bad 3.0 |
| Ours w/o CT. | 0.38 | 4.24 | 0.34 | 2.13 | 56.34 | 15.23 |
| fgsm 20-epoch | 0.44 | 6.97 | 0.65 | 0.86 | 23.92 | 2.46 |
| adv-3 20-epoch | 0.45 | 7.67 | 0.72 | 0.79 | 20.91 | 1.91 |
| adv-9 20-epoch | 0.52 | 10.36 | 1.06 | 0.79 | 21.50 | 2.18 |
| adv-9 50-epoch | 0.52 | 10.36 | 1.06 | 0.74 | 19.26 | 1.98 |
| Ours | **0.36** | **3.61** | **0.27** | 0.88 | 21.20 | 3.75 |
| Ours + adv. | 0.41 | 5.77 | 0.52 | **0.61** | **13.46** | **1.39** |

Table 9: Stereo-constrained attack results under different adversarial training settings. The first 5 rows show the results of our modified version with two feature backbones.

## C.3 Ablation Study on Adversarial Training

Here we modify our model by using two feature backbones while keeping other components fixed and use this setting to study the effects of different PGD iteration steps and training epochs on adversarial training in Table 9. Our method is not suitable for justifying these hyper-parameters since it has significantly stronger robustness.

In this experiment, we test the modified counterpart under FGSM attack, the unconstrained PGD attacks with different iterations (3 and 9) and training epochs (20 and 50). In order for adversarial training to be effective, the attack should be as strong as possible [3]. Table 9 shows that all the model trained with PGD attacks are indeed more robust than the one with FGSM attack. However, it does not make much difference for using 9 iterations or 50 epochs, showing that 3 iterations with 20 epochs are sufficient for the adversarial training.

Besides justifying the hyper-parameters, this result also shows that our method is indeed more robust than its counterpart as it uses the proposed multi-scale Census Transform for the matching.

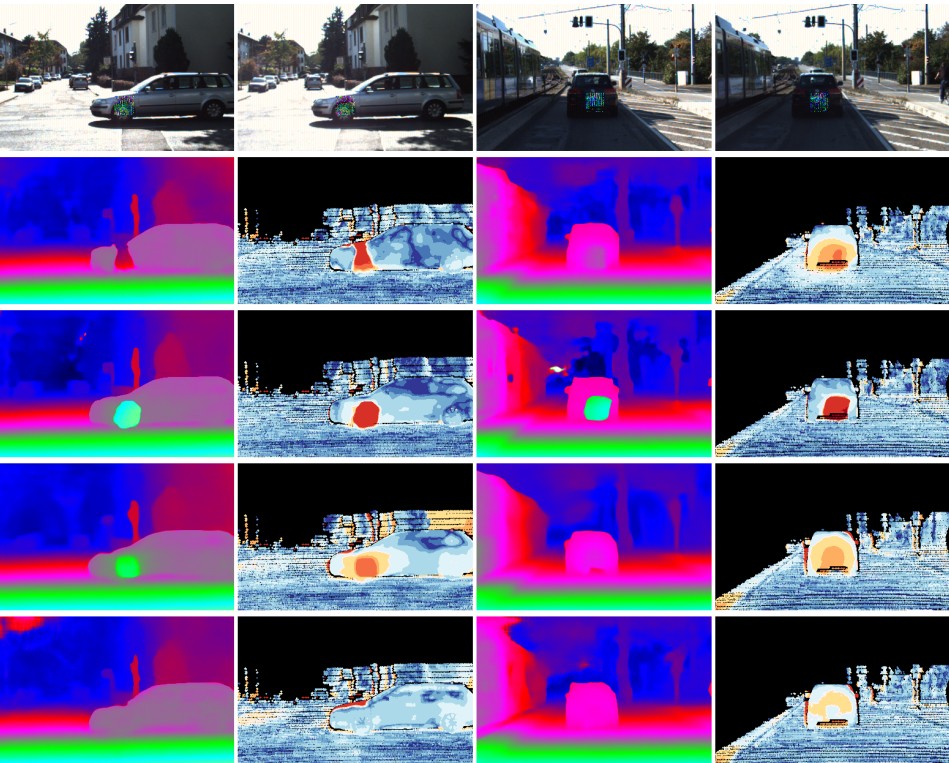

Figure 10: Illustration of the adversarial patch attack (3/5). The first row shows the attacked image pairs. PSMNet [9], GANet [10], LEAStereo [11], and our method are shown on the second, third, fourth, and fifth row respectively. The first and the third columns are the after attack disparity maps, and the second and the fourth columns are the after attack error maps from the ground truth.

| | Clean | | | Synthetic Patch Attack | | |
|---|---|---|---|---|---|---|
| Models | EPE | Bad 1.0 | Bad 3.0 | EPE | Bad 1.0 | Bad 3.0 |
| PSMNet | 0.28 | 2.30 | 0.21 | 0.80 | 15.41 | 4.46 |
| GANet-Deep | **0.25** | **1.46** | **0.12** | 1.84 | 12.52 | 9.09 |
| LEAStereo | 0.41 | 5.60 | 0.69 | 1.34 | 20.32 | 6.18 |
| Ours | 0.40 | 4.79 | 0.46 | 0.54 | 7.58 | 0.85 |
| Ours w/o ctx. | 0.46 | 5.51 | 0.63 | **0.48** | **6.42** | **0.81** |
| PSMNet + adv. | 0.58 | 12.56 | 1.44 | 1.44 | 26.55 | 8.59 |
| GANet + adv. | 0.55 | 11.24 | 1.32 | 2.31 | 23.5 | 11.98 |
| LEAStereo + adv. | 0.71 | 15.87 | 2.24 | 1.96 | 30.88 | 11.72 |
| Ours w/o ctx. + adv. | **0.52** | **8.95** | **1.03** | 0.54 | 9.45 | **1.18** |
| Ours + adv. | 0.53 | 11.25 | 1.17 | 0.58 | 12.01 | 1.40 |

Table 10: Adversarial Patch Attack Results in the KITTI2015 training dataset with photometric consistency retained in attack.

## C.4   Supplementary Results on the Experiments

Here we provide the EPE and Bad 1.0 for the adversarial patch attack and the transferability of adversarial robustness experiments. From Tables 10 and 11, our method shows significantly stronger robustness also in EPE and Bad 1.0, especially in Bad 1.0.

In addition, we show two different versions of the unconstrained attacks in KITTI2015, where one use the ground truth to attack, and the other use the original neural network prediction. Results are shown in Table 12. It shows that using prediction is weaker than using the ground truth for unconstrained attacks.

## C.5   Occluded Regions

For the proposed stereo-constrained PGD attack, we disallow to attack and evaluate occluded regions of the reference image, which prevents the perturbation to attack the regions where the estimation

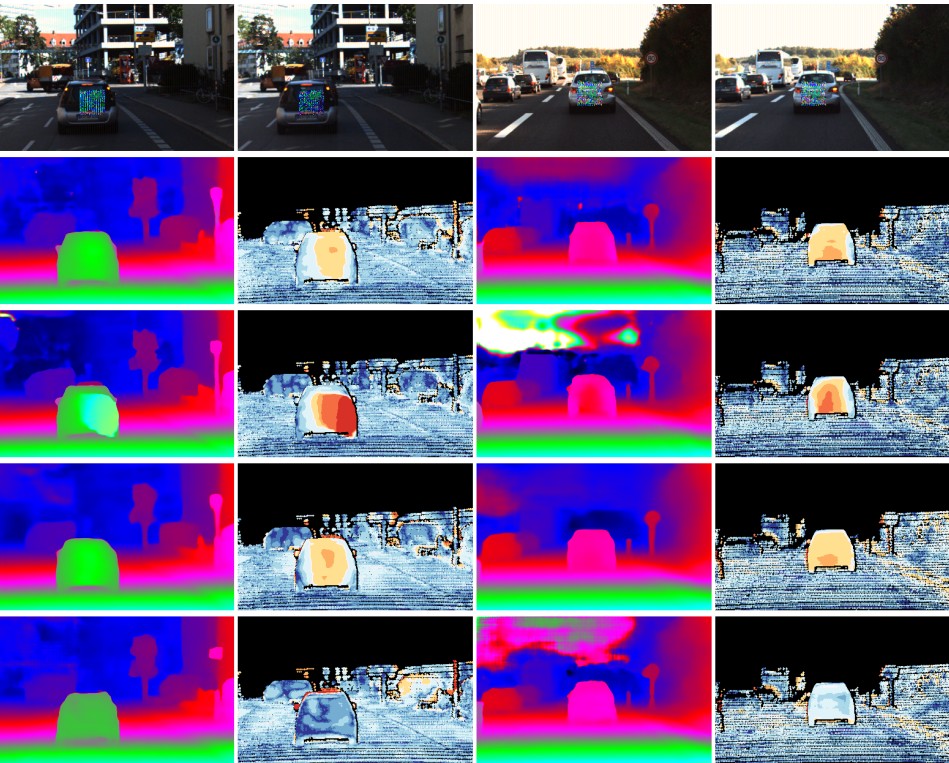

Figure 11: Illustration of the adversarial patch attack (4/5). The first row shows the attacked image pairs. PSMNet [9], GANet [10], LEAStereo [11], and our method are shown on the second, third, fourth, and fifth row respectively. The first and the third columns are the after attack disparity maps, and the second and the fourth columns are the after attack error maps from the ground truth.

| Models | Clean | | | PGD Attack ($\epsilon = 0.03$) | | | PGD Attack ($\epsilon = 0.06$) | | |
|---|---|---|---|---|---|---|---|---|---|
| | EPE | Bad 1.0 | Bad 3.0 | EPE | Bad 1.0 | Bad 3.0 | EPE | Bad 1.0 | Bad 3.0 |
| PSMNet + adv. | 0.8 | 18.24 | 3.03 | 1.37 | 33.87 | 6.72 | 2.07 | 51.00 | 12.28 |
| GANet + adv. | 0.80 | 18.82 | 3.03 | 1.45 | 36.79 | 7.75 | 2.29 | 54.07 | 15.42 |
| LEAStereo + adv. | 0.85 | 20.17 | 3.56 | 1.48 | 38.24 | 8.60 | 2.3 | 55.67 | 16.24 |
| Ours w/o ctx. | 0.49 | 6.40 | 1.23 | 1.58 | 26.69 | 9.66 | 1.92 | 32.52 | 12.87 |
| Ours | 0.46 | 6.07 | 1.00 | 1.37 | 22.45 | 7.29 | 1.80 | 29.61 | 10.50 |
| Ours w/o ctx. + adv. | 0.65 | 11.18 | 2.25 | 1.28 | 24.62 | 6.64 | 1.48 | 28.93 | 8.31 |
| Ours + adv. | **0.56** | **10.14** | **1.73** | **0.89** | **18.35** | **4.14** | **1.09** | **22.64** | **5.95** |

Table 11: Transferrability of Adversarial Robustness: stereo-constrained 20-step PGD Attack Results in the KITTI2012 training dataset using adversarially trained neural networks on KITTI2015.

does not rely on matching. Nonetheless, it is still possible to make perturbation on the occluded regions of the right image to hinder the matching, e.g. by creating false positive correspondence. We also consider this situation and experiment with *an even weaker attack* such that the occluded regions of both the left and right images will not be attacked.

The results are shown in Table 13 Table 15, which are consistent with those reported in the submission. Our methods are the best in all metrics except for EPE in ScenFlow with $\epsilon = 0.03$.

## C.6  Results in SceneFlow

We first compare the adversarial robustness. Table 14 and Table 15 shows the comparisons: **our models are much more robust than the prior art**. As the original SceneFlow dataset does not have ground truth occlusion, we use a subset provided by the same authors [5] with occlusions. We randomly select $1,000$ images from the test data. Our methods show significantly better robustness against attacks. In fact, our methods are the best in all metrics except for EPE in ScenFlow with $\epsilon = 0.03$ in Table 15. We note that if we do not allow to use the proposed differentiable approximation

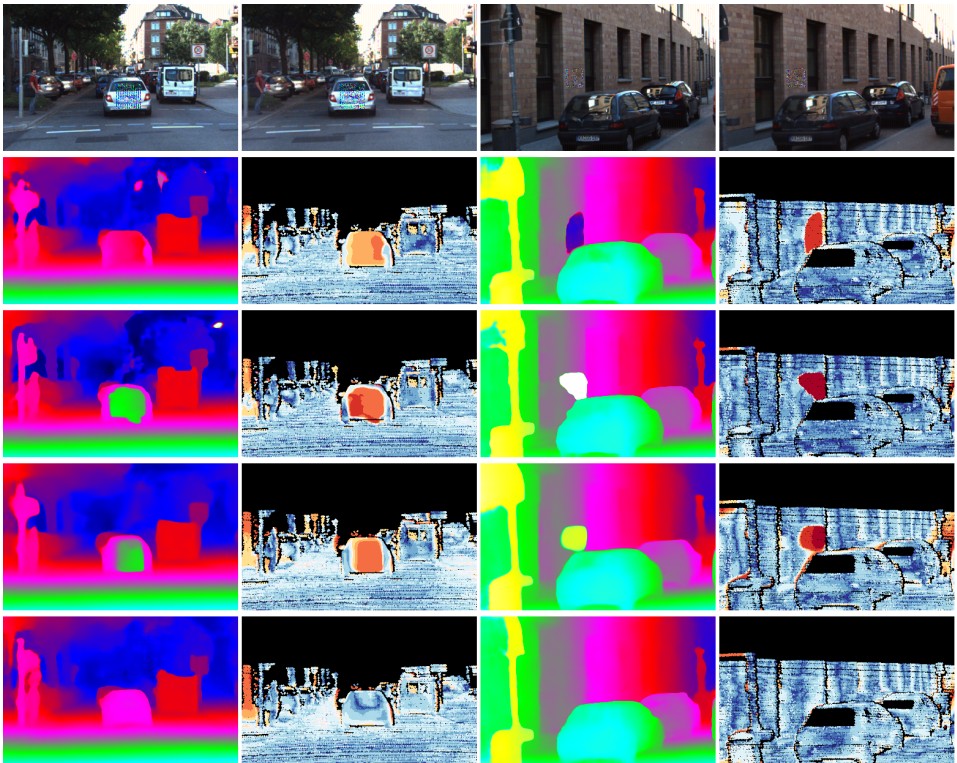

Figure 12: Illustration of the adversarial patch attack (5/5). The first row shows the attacked image pairs. PSMNet [9], GANet [10], LEAStereo [11], and our method are shown on the second, third, fourth, and fifth row respectively. The first and the third columns are the after attack disparity maps, and the second and the fourth columns are the after attack error maps from the ground truth.

| Models | PGD Attack ($\epsilon = 0.03$) w/ GT | | | PGD Attack ($\epsilon = 0.03$) w/ Prediction | | |
|---|---|---|---|---|---|---|
| | EPE [px] | Bad 1.0 [%] | Bad 3.0 [%] | EPE [px] | Bad 1.0 [%] | Bad 3.0 [%] |
| PSMNet | 91.08 | 92.75 | 89.91 | 58.75 | 84.11 | 70.49 |
| GANet | 23.75 | 89.48 | 79.11 | 19.49 | 68.01 | 51.58 |
| LEAStereo | 14.71 | 82.42 | 64.31 | 12.03 | 74.21 | 54.00 |
| Ours w/o ctx. | 2.36 | 41.34 | 16.30 | 2.27 | 34.98 | 14.74 |
| Ours | **1.81** | **36.42** | **11.29** | **1.53** | **29.25** | **10.09** |

Table 12: Vanilla 20-step PGD Attack Results in the KITTI2015 training dataset [6]. The PGD attacks are learned using either the GT disparity map or the predicted disparity map from clean images in the loss function used to compute PGD. The performance are still measured in terms of the GT disparity map.

of Census Transform, our method shows much better robustness, thanks to the non-differentiable cost volume computation.

We also compare the results using the entire image, instead of only non-occlusion regions in evaluating attack performance. Table 16 shows the comparisons. Our method obtains competitive performance against state-of-the-art methods. Recent work suggests there exists an inherent conflict between accuracy and robustness [37, 38]. From this perspective, the comparable performance on clean images and the significantly better robustness of our method show that the proposed design for stereo matching is effective.

# D   Environment

All the experiments were done on a Nvidia DGX server running Ubuntu 18.04.5, which equips 4 Tesla V-100 GPUs, each has 32 gigabytes of memory. For Pytorch, we use version 1.8.0 with CUDA 11.1.

| Models | Clean | | | PGD Attack ($\epsilon = 0.03$) | | | PGD Attack ($\epsilon = 0.06$) | | |
|---|---|---|---|---|---|---|---|---|---|
| | EPE | Bad 1.0 | Bad 3.0 | EPE | Bad 1.0 | Bad 3.0 | EPE | Bad 1.0 | Bad 3.0 |
| PSMNet | 0.28 | 2.00 | 0.16 | 1.29 | 45.79 | 5.12 | 5.19 | 70.73 | 30.08 |
| GANet | **0.25** | **1.42** | **0.10** | 1.25 | 39.76 | 5.92 | 2.93 | 63.88 | 25.84 |
| LEAStereo | 0.37 | 4.54 | 0.42 | 1.44 | 46.57 | 7.61 | 3.20 | 66.26 | 27.45 |
| Ours w/o ctx. | 0.38 | 4.14 | 0.32 | 0.60 | 12.95 | 1.16 | 0.64 | 14.39 | 1.42 |
| Ours | 0.36 | 3.61 | 0.27 | **0.53** | **9.90** | **0.75** | **0.59** | **11.97** | **0.93** |

Table 13: Stereo-Constrained 20-step PGD Attack Results in the KITTI2015 training dataset [6]. Attacks on occluded regions of both the left and the right image are disallowed.

| Models | Clean (Non-occlusion regions) | | | After PGD Attack ($\epsilon = 0.03$) | | | After PGD Attack ($\epsilon = 0.06$) | | |
|---|---|---|---|---|---|---|---|---|---|
| | EPE | Bad 1.0 | Bad 3.0 | EPE | Bad 1.0 | Bad 3.0 | EPE | Bad 1.0 | Bad 3.0 |
| PSMNet | 1.56 | 20.62 | 5.36 | 12.37 | 68.12 | 30.75 | 25.51 | 75.66 | 51.13 |
| GANet | 1.04 | 10.82 | 3.37 | 11.52 | 51.47 | 36.10 | 28.54 | 72.75 | 64.03 |
| LEAStereo | 1.03 | 8.87 | **2.69** | 12.12 | 55.69 | 33.82 | 22.30 | 68.04 | 53.63 |
| Ours | **1.02** | **8.85** | 3.28 | 9.87 | 31.94 | 25.67 | 20.74 | 45.84 | 41.37 |
| Ours w/o ctx. | 1.16 | 9.49 | 3.55 | **9.23** | **30.61** | **25.15** | **10.85** | **32.88** | **27.84** |
| Ours† | 1.02 | 8.85 | 3.28 | 2.25 | 12.64 | 6.71 | 5.79 | 18.09 | 13.04 |

Table 14: Stereo-Constrained 20-step PGD Attack Results in SceneFlow [5]. † shows results by our method without using the modified census transform in learning attacks, which are much more resistant to attacks. See text for detail.

| Models | Clean | | | After PGD Attack ($\epsilon = 0.03$) | | | After PGD Attack ($\epsilon = 0.06$) | | |
|---|---|---|---|---|---|---|---|---|---|
| | EPE | Bad 1.0 | Bad 3.0 | EPE | Bad 1.0 | Bad 3.0 | EPE | Bad 1.0 | Bad 3.0 |
| PSMNet | 1.56 | 20.62 | 5.36 | 12.32 | 68.15 | 30.76 | 20.15 | 74.50 | 46.74 |
| GANet | 1.04 | 10.82 | 3.37 | 14.37 | 57.90 | 42.03 | 23.59 | 69.28 | 57.90 |
| LEAStereo | 1.03 | 8.87 | **2.69** | **9.32** | 54.86 | 31.16 | 16.51 | 64.26 | 45.77 |
| Ours w/o ctx. | 1.16 | 9.49 | 3.55 | 11.63 | **34.18** | **29.26** | **13.09** | **36.55** | **32.21** |
| Ours | **1.02** | **8.85** | 3.28 | 11.70 | 35.36 | 29.56 | 16.08 | 42.14 | 37.21 |

Table 15: Stereo-Constrained 20-step PGD Attack Results in SceneFlow [5]. Attacks on occluded regions of both the left and the right image are disallowed.

| Models | Params [M] | EPE [px] | Bad 1.0 [%] | Bad 3.0 [%] |
|---|---|---|---|---|
| PSMNet | 3.5M | 1.49 | 20.6 | 5.9 |
| GANet | 6.6M | **0.82** | 9.0 | 3.5 |
| LEAStereo | 1.8M | 0.83 | **8.0** | **3.3** |
| Ours w/o ctx. | 1.9M | 1.10 | 9.7 | 4.4 |
| Ours | 2.7M | 0.84 | 8.8 | 3.7 |

Table 16: Result comparisons using clean images in the SceneFlow dataset [5].