# OpenReview forum: "Revisiting Non-Parametric Matching Cost Volumes for  Robust and Generalizable Stereo Matching"
_NeurIPS.cc/2022/Conference — NeurIPS 2022 Accept_

### Official Review · Reviewer_HFdJ · 2022-07-10

**Rating:** 4
**Confidence:** 5
**Soundness:** 3 good
**Presentation:** 3 good
**Contribution:** 2 fair

**Summary:**

The paper presented a new design for stereo matching, which utilizes DNNs to aggregate/optimize non-parametric cost volumes with parametric contextual features. It harnesses the best of classic features (multi-scale census transform) and end-to-end trainable DNNs for adversarially-robust and cross-domain generalizable stereo matching. The proposed method is motivated by the observation that DNN-based stereo matching methods can be deceived by a type of physically realizable attacks that entail stereo constraints in learning the perturbation.

**Questions:**

- Please clarify the differences with existing attack methods;

- Please discuss the missing references in analyzing stereo matching.

**Limitations:**

The authors partially addressed the limitations. More analysis should be provided.

**Strengths And Weaknesses:**

Strengths:

+ The paper proposes a new design for stereo matching by utilizing DNNs to aggregate/optimize non-parametric cost volumes with parametric contextual features, which shows significantly better adversarial robustness and improved cross-domain (Sim2Real) generalizability when no-fine tuning is used.

+ It presents the stereo-constrained projected gradient descent (PGD) attack method, which by design preserves photometric consistency to show the more serious vulnerabilities of state-of-the-art DNN-based stereo matching methods.

Weaknesses:

- The cost aggregation problem perspective has been widely exploited.

- The weak performance on the KITTI 2015 leaderborder.

- Other related work in attacking stereo matching networks.

--- Open-world stereo video matching with deep rnn, ECCV 2018.

---

> ### Author Response · Authors · 2022-08-02
> **Response 1/1**
>
> Thank you for your time reviewing our paper. In the following, we address it point by point.
>
> **Comments**: Other related work in attacking stereo matching networks.
>
> **Response**: Currently, to our knowledge, the only work that also studies attacking stereo matching networks is [26]. We discuss the difference between our work and [26] in details in the related work section. They study unconstrained adversarial attacks on both images separately. However, without enforcing photometric consistency, these attacks will violate the underlying physical properties of binocular vision and thus are not realizable in practice. For example, their proposed attacks cannot compute adversarial patches to fool stereo systems. More importantly, they did not propose new network architecture to defend, which is the main contribution in this work.
>
>
> **Comments**: The cost aggregation problem perspective has been widely exploited.
>
> **Response**: The design of combining a non-parametric cost volume and a reference-image only parametric context extractor is unique. While there are other networks, e.g. SGM-Net and GANet, that try to embed the form of SGM in the cost aggregation stage, our method does not enforce a specific form of the cost aggregation and thus follows a different design philosophy. It is also important that we use a context extractor to aid the cost aggregation, which was never exploited before. In experiments, we show that this design strategy is effective in achieving both high robustness and generalizability simultaneously.
>
>
> **Comments**: The weak performance on the KITTI 2015 leaderborder.
>
> **Response**: As shown in [2], there is a trade-off between accuracy and adversarial robustness in DNNs, thus there will be drops in accuracy in adversarial robust models. By utilizing the binocular property of stereo matching, our method has a much smaller accuracy gap from the current best method than other tasks, such as image classification. As shown in [28], the top-1 accuracy of the most effective adversarial training method for ResNet-152 on ImageNet drops from 78\% to 67\% on clean images, and a typical adversarial training often only maintains a 57\% accuracy. Jointly maintaining high accuracy and adversarial robustness is very difficult and there are no models in ImageNet that can achieve such a high accuracy and robustness even with the help of adversarial training, not to mention the built-in robustness of a model. Furthermore, our model also has a very competitive accuracy on SceneFlow, as shown in Talbe 18.
>
>
> **Comments**: Open-world stereo video matching with deep rnn, ECCV 2018.
>
> **Response**: Thank you for pointing out this interesting work and we will discuss it in the related work section. This work considers stereo videos as input and it is different from our current setting of stereo matching using only a pair of images.

---

### Official Review · Reviewer_RUGj · 2022-07-11

**Rating:** 6
**Confidence:** 5
**Soundness:** 3 good
**Presentation:** 3 good
**Contribution:** 3 good

**Summary:**

The paper addresses robustness to adversarial attacks and generalizability in stereo matching via a network architecture that operates on a cost volume generated by a multi-scale Census Transform (CT), instead of learned image features. This cost volume has superior robustness and generalization properties compared to recent state-of-the-art stereo matching networks most likely because it is the input to a network that only aggregates information and is less susceptible to shortcuts. Experiments on several datasets using three strong baseline algorithms are presented in the paper.

**Questions:**

I am most interested in the authors’ opinion on how the context extractor does not over-specialize to the training data.

Please respond to the technical comments briefly. Clarifying the quantitative results is important.


**Limitations:**

Limitations are discussed briefly, which is sufficient due to the nature of the paper.

**Strengths And Weaknesses:**

Strengths:

The proposed approach is successful in both aims: it shows robustness to image intensity perturbation attacks as well as strong results in SimToReal transfer.

The paper provides useful insights on the behavior of modern network architectures, their surprising failures in the presence of stereo-constrained attacks and feature extraction and matching, rather than cost aggregation, being the cause of the vulnerability of DNNs applied outside their training domains.

The stereo-constrained projected gradient descent (PGD) attack method is an additional contribution, but I consider it less important than the above.

Experimental validation is extensive, and the proposed network achieves high accuracy on several datasets. See the technical comments below, however.


Weaknesses:

An approach for stereo matching relying on hand-crafted features aggregated by neural networks has been published by:
C. Cai, M. Poggi, S. Mattoccia, and P. Mordohai. Matching-space Stereo Networks for Cross-domain Generalization. 3DV 2020.
It relies on multiple hand-crafted features, including CT, and shows good generalization and SimToReal performance.

Recent papers on generalization in stereo matching include the following:
F. Zhang, X. Qi, R. Yang, V. Prisacariu, B. Wah, and P. Torr. Domain-invariant stereo matching networks. ECCV, 2020.
Zhengfa Liang, Yulan Guo, Yiliu Feng, Wei Chen, Linbo Qiao, Li Zhou, Jianfeng Zhang, and Hengzhu Liu. Stereo matching using multi-level cost volume and multi-scale feature constancy. TPAMI, 2019.
Zhelun Shen, Yuchao Dai, and Zhibo Rao. CFNet: Cascade and fused cost volume for robust stereo matching. CVPR 2021.

I find the role and effectiveness of the context extractor perplexing. It seems to contradict the main argument of the paper that the feature extractors, typically a Siamese network, used to construct the cost volume cause overfitting, presumably by learning non-essential features of the training set. (The last phrase is mine, not taken from the paper.) I wonder what protects the context extractor, which is only applied on the left image, from suffering from the same limitation. I speculate that such a single-image network would associate appearance with depth in ways that do not transfer across datasets.

Sections 3.2 and 4.4 contain arguments that the advantage of CT is that it is not differentiable and this prevents gradients from propagating through it. Cleary, gradient propagation can be stopped before any operation whether it is differentiable or not. Considering certain parts of the system unlearnable (freezing them during training) is the key characteristic.


Technical Comments:

I consider the following comments relatively easy to correct or clarify. I will not repeat them in the Questions section of the review.

Disparity is typically defined as the difference between the left and right x-coordinates, i.e. d= x_L – x_R. Lines 50 and (1) are based on the opposite definition.

127: It is not clear to me that not perturbing occluded pixels is the correct approach.

133-135: this is only true if the patches have constant disparity.

209-211: are even-sized windows also used? Where is the reference pixel placed within the window?

Table 3: the Middlebury dataset does not make a Bad3.0 error available. What is actually shown in the last column?

Table 4: the Bad3.0 results on clean images are way better than anything reported in the literature and the official leaderboard. (Several error rates below 1% are shown, including a minimum of 0.22%. The top submission to the benchmark is at 1.49% on unoccluded pixels.) I am aware that these are results on the training set, but the degree of difficulty of the training set is comparable to that of the test set. Is it possible that the networks have seen these images? The same is true for the 1.00% Bad3.0 error on KITTI 2012 reported in Table 6.

Table 8 contains reasonable results. It is not surprising that the cost of robustness and generalization is somewhat reduced in-domain accuracy. There are tradeoffs between generalization and specialization. (This is a neutral comment, placed here to complement the comments above.)

Section 5, among other topics, discusses physically realizable attacks on stereo matching in the context of autonomous driving. I am not convinced that such attacks are feasible over sequences of frames captured by mobile cameras. Attacks in the form of specular or textureless surfaces may be more dangerous to most current stereo matching systems.


Minor Comments

12: “due to the data hungry of DNNs” should be corrected since hungry is not a noun.

27, (3): the cross-product symbol should not be used for scalar multiplication.

41-45: concatenation is one of the options for constructing the cost volume.

48: “learn to perform matching”

51: delete “in” before “regardless”

177: “the performance drop on clean images” seems wrong.

393: provide correct publication venue.

Checklist 4(b): the licenses of the used assets (datasets and network implementations) are not mentioned.

---

> ### Author Response · Authors · 2022-08-02
> **Response 1/2**
>
> Thank you for your time reviewing our paper. In the following, we address it point by point.
>
> **Comments**: The role and effectiveness of the context extractor.
>
> **Response**: Thank you for pointing this out. We will explain this in more details in revision. We also consider this as a surprising and important finding in stereo matching because the usage of context extractor improves accuracy while maintaining robustness. While the matching component alone can explain away most of the non-occluded texture-rich regions, training the context extractor together with the matching component can encourage the context extractor to focus on context cues for regions where matching alone suffers, such as occluded, textureless or repetitive regions. To attack the matching component and the context extractor at the same time, an attaching method (such as the PGD used in the submission) must find ways to alter both the cost volume and the context cues from the reference image. The fact that the robustness is not affected significantly shows that the alternation of the context cues either contradicts the alternation of the matching component, or being too weak compared to the matching component. In other words, the gradients from the context extractor have different directions or much smaller magnitudes compared to the gradients from the matching component. This finding shows a way to improve accuracy without sacrificing robustness in stereo matching.
>
> The context cues from the reference image is dataset dependent since it learns from the semantic information of the data. Compared to the gain of accuracy within the same domain (as shown in Table 18), we think the drop of Sim2Real accuracy is reasonable (as shown in Table 3). Also, the context extractor works well for datasets that have similar domains (such as from KITTI2015 to KITTI2012 as shown in the Clean section in Table 6).
>
>
> **Comments**: Sections 3.2 and 4.4 contain arguments that the advantage of CT is that it is not differentiable and this prevents gradients from propagating through it. Cleary, gradient propagation can be stopped before any operation whether it is differentiable or not. Considering certain parts of the system unlearnable (freezing them during training) is the key characteristic.
>
> **Response**: This paper considers the white-box adversarial attack setting in which an attacker has the full access to both the model architecture and learned parameters. Even the model was trained with certain unlearnable parts, disabling the back propagation during training the model won't stop the attacker to bring the back propagation back and compute the adversarial patch, thus creating an illusion of safety. This can be extended to even truly non-differentiable modules as shown in the obfuscated gradient problem [8], where the authors show that they can approximate the gradients for most non-differentiable modules and still be able to attack the models. Therefore, in section 3.2, we provide a differentiable approximation method for the comparison operator and show the surprising result that this model is still attackable, though being much more robust than others. Note that this result is non-trivial as the attack needs to alter an image pixel to pass certain threshold in order to change the output of the census transform, even when the gradients are provided. This means that for the $\epsilon$-bounded attack, only pixels with similar colors within the patch will be altered.
>
> On the other hand, the accuracy of a truly adversarial robust model will not drop much even if the strongest adversarial patch is applied. It is our goal to develop such a model. We will revise the arguments on the advantages of CT in sections 3.2 and 4.4 to be more precise.
>
>
> **Comments**: 127: It is not clear to me that not perturbing occluded pixels is the correct approach.
>
> **Response**: By excluding occluded pixels, our goal is to show that the matching component is still vulnerable even with this constraint, and thus seeking more robust matching components is a pressing need in robust stereo matching. In the meanwhile, without excluding occluded pixels, the attack can potentially alter the occluded pixels to have similar colors as other pixels on the right image, thus creating false positive matches. We also provide experiments on unconstrained attacks, showing that our model is still much more robust when occluded pixels can be attacked (see Table 4).
>
> **Comments**: 133-135: this is only true if the patches have constant disparity.
>
> **Response**: Thank you for pointing this out. We will rephrase it more clearly.
>
>
> **Comments**: 209-211: are even-sized windows also used? Where is the reference pixel placed within the window?
>
> **Response**: Yes, we also use even-sized windows. The reference pixel is placed at the ``center" of the kernel which is determined in the same way as the 2D convolution with an even-sized kernel implemented in PyTorch.

---

> > ### Author Response · Authors · 2022-08-02
> > **Response 2/2**
> >
> > **Comments**: Table 3: the Middlebury dataset does not make a Bad3.0 error available. What is actually shown in the last column?
> >
> > **Response**: This tables shows comparisons for the Sim2Real cross-domain generalizability from the SceneFlow trained models to the KITTI 2015, KITTI 2012 and Middleburry training datasets without any fine-tuning. We calcuate the Bad3.0 in Middleburry training dataset following the metric protocol for each model.
> >
> > **Comments**: Table 4: the Bad3.0 results on clean images are way better than anything reported in the literature and the official leaderboard. (Several error rates below 1\% are shown, including a minimum of 0.22\%. The top submission to the benchmark is at 1.49\% on unoccluded pixels.) I am aware that these are results on the training set, but the degree of difficulty of the training set is comparable to that of the test set. Is it possible that the networks have seen these images? The same is true for the 1.00\% Bad3.0 error on KITTI 2012 reported in Table 6.
> >
> > **Response**: Table 4 shows the evaluation results on the whole train-validation set (i.e. 200 images), so the clean error rates are very low. The reason that the whole train-validation set is used lies in two-fold: (i) There is no official train-validation split available, and different methods adopt different settings (e.g., 140-180 images out of the total 200 images are used as training images by different methods). (ii) The clean results in Table 4 are mainly for the purpose of showing the accuracy drop before and after attacks. Therefore, we also use the whole training set instead of the validation set for more fair comparison.
> > However, in Table 6, all networks were trained in KITTI2015. Thus the clean results are more meaningful because they show the cross-domain accuracy from KITTI2015 to KITTI2012. Our method indeed outperforms others in this regard.
> >
> >
> > **Comments**: Section 5, among other topics, discusses physically realizable attacks on stereo matching in the context of autonomous driving. I am not convinced that such attacks are feasible over sequences of frames captured by mobile cameras. Attacks in the form of specular or textureless surfaces may be more dangerous to most current stereo matching systems.}
> >
> > **Response**: In general, we agree that the proposed attacks may not be always feasible over sequences of frames captured by mobile cameras. There exist certain common scenarios in autonomous driving in which the proposed attacks are indeed physically feasible, e.g. vehicles are co-moving with similar speeds on highways. Furthermore, a computed adversarial patch has certain transferability over different depths as shown in Figure 7. However, whether we can use temporal information to defend is another topic. In fact, current optical flow is very vulnerable -- even a static placed adversarial patch can easily alter the prediction [22]. Therefore, we think our approach is important for the safety and robustness of the current stereo systems.
> >
> > We also want to emphasize the difference between our approach and attacking two images separately without any constraints. The latter is not physically realizable because it violates the photometric constraint whereas our approach can compute an adversarial patch and stick it on the car to actually attack the current stereo system (as shown in Figures 8-12).

---

> > > ### Comment · Reviewer_RUGj · 2022-08-07
> > > **Response to Authors**
> > >
> > > I have now read all reviews and author responses. I will focus on the response to my review in this posting without repeating issues that I consider closed after the authors’ response.
> > >
> > > *Related work.* I realize now that I did not list discussing the references I suggested as a requirement for the authors’ response, but a few lines on them would be useful.
> > >
> > > *The role and effectiveness of the context extractor.* This is an interesting finding. I am more interested in robustness than resilience against attacks. I understand how it helps in the latter. The authors’ perspective that the context extractor contributes to occluded, textureless and repetitive regions is interesting and worth discussing in more detail if the paper is accepted. The limitation that it introduces dependence on the training domain should be included in the discussion.
> > >
> > > *Blocking gradient backpropagation.* I consider the response satisfactory, even though in the white box case more attacks are possible.

---

> > > > ### Author Response · Authors · 2022-08-07
> > > > **Response to Reviewer RUGj**
> > > >
> > > > Thank you very much for looking into our responses, and for your further valuable comments.
> > > >
> > > > *Related work*
> > > >
> > > > **Comments**:  An approach for stereo matching relying on hand-crafted features aggregated by neural networks has been published by: C. Cai, M. Poggi, S. Mattoccia, and P. Mordohai. Matching-space Stereo Networks for Cross-domain Generalization. 3DV 2020. It relies on multiple hand-crafted features, including CT, and shows good generalization and SimToReal performance.
> > > >
> > > > **Response**: Thank you for pointing out this interesting work and we will discuss it more in the revision. It is similar to both [19] and our method in the spirit of bring the non-parametric cost volume back for stereo matching. Our approach is different that it also utilizes a parametric context extractor, which improves the in-domain accuracy. From the adversarial robustness perspective,  our approach only uses the Census Transform (CT) and shows that CT only is sufficiently effective without combining it combining it with SAD or ZNCC, as the ablation studies shown in Table 7 in the submission. Overall, revisiting non-parameteric cost volume shows good advantages for stereo matching, especially combining with the context extractor.
> > > >
> > > > **Comments**: Recent papers on generalization in stereo matching include the following: F. Zhang, X. Qi, R. Yang, V. Prisacariu, B. Wah, and P. Torr. Domain-invariant stereo matching networks. ECCV, 2020. Zhengfa Liang, Yulan Guo, Yiliu Feng, Wei Chen, Linbo Qiao, Li Zhou, Jianfeng Zhang, and Hengzhu Liu. Stereo matching using multi-level cost volume and multi-scale feature constancy. TPAMI, 2019. Zhelun Shen, Yuchao Dai, and Zhibo Rao. CFNet: Cascade and fused cost volume for robust stereo matching. CVPR 2021.
> > > >
> > > > **Response**: Thank you for suggesting these papers, we will include them in revision and discuss more from the cross-domain generalization perspective.
> > > >
> > > > In domain-invariant stereo matching networks by Zhang et al, they propose a “domain normalization” approach that regularizes the distribution of learned representations to allow them to be invariant to domain differences.
> > > >
> > > > In stereo matching using multi-level cost volume and multi-scale feature constancy by Liang et al, they use the initial estimation to refine by encouraging multi-scale feature constancy, which is introduced to measure the correctness of the initial disparity in feature space.
> > > >
> > > > In the CFNet: Cascade and fused cost volume for robust stereo matching by Shen, Dai and Rao, they propose to fuse cost volume representation using multiple low-resolution dense cost volumes and employ an iterative variance-based uncertainty estimation to adjust the next stage disparity search space and progressively prune out the space of unlikely correspondences.
> > > >
> > > > While these works also improve cross-domain generalizabiity, our approach is orthogonal to theirs (i.e. different normalization techniques and cost volume refinement strategies) and we believe they can be used together to further improve the cross-domain accuracy. From the perspective of adversarial attacks, the first two approaches focus on the cross-domain representation power of DNN features, which will potentially make them more vulnerable against attacks.
> > > >
> > > > *The role and effectiveness of the context extractor*
> > > >
> > > > **Comments**: The authors’ perspective that the context extractor contributes to occluded, textureless and repetitive regions is interesting and worth discussing in more detail if the paper is accepted. The limitation that it introduces dependence on the training domain should be included in the discussion.
> > > >
> > > > **Response**: Thank you. We will show qualitative results on how the model w/ context extractor contributes to occluded, textureless and repetitive regions, compared to the model w/o context extractor.  We will include the discussions on the domain dependence introduced by the context extractor in revision.

---

### Official Review · Reviewer_WPxi · 2022-07-11

**Rating:** 7
**Confidence:** 3
**Soundness:** 3 good
**Presentation:** 3 good
**Contribution:** 3 good

**Summary:**

The paper considers the problem of adversarial on binocular stereo matching systems. It shows that an adversarial attack based on PGD that is photometrically consistent and consistent with stereo constraints significantly affects the performance of learned features for stereo matching. To counter the attack, the paper proposes to not rely on learned features but rather on handcrafted features (here, the census transform is used). In order to compensate for using weaker features, the cost aggregation stage is learned (and optionally, a network is used to provide higher-level context). The paper shows that this approach has multiple advantages: (1) as low-level features are fixed and not learned, learning focuses on the higher-level task of identifying valid structures in the cost volume, it generalizes better from training purely on synthetic data to being used on real data. (2) the proposed approach is more robust against the proposed adversarial attacks. In addition, when trained with adversarial training, the drop in performance on unaltered images is significantly reduced. The paper concludes that current networks for learning features for stereo matching do not properly learn to match.

**Questions:**

Please address the weaknesses listed above (and in particular W2) in a potential rebuttal.

**Limitations:**

As detailed above under W1, the limitations of the proposed approach to counter the adversarial attack are not fully discussed.

**Strengths And Weaknesses:**

Strengths:
S1) The paper tackles an interesting problem of practical relevance, namely robustness to adversarial attacks. Furthermore, it asks an interesting theoretical questions (do matching networks really learn to match?) and aims to answer this using adversarial attacks.

S2) The proposed adversarial attack is technically sound and its effectiveness is shown by evaluating multiple state-of-the-art baselines on modified images. Furthermore, the paper argues convincingly that the proposed attack is feasible in the real world (although the drop in performance is significantly smaller in the scenario considered in the paper).

S3) The proposed counter to the attack is well-described and intuitively meaningful. Detailed experiments show that it performs well on standard benchmark datasets.

S4) The paper is overall easy to read and to follow and tells a consistent and interesting story.

Weaknesses:
W1) The paper focuses on a variant of the stereo matching problem where both images are taken at the same time (from relatively similar viewpoints), e.g., binocular stereo for self-driving cars. In this setting, the assumption of "strong photometric consistency between stereo images" holds. There are other stereo scenarios, e.g., as part of multi-view stereo algorithms, where there can be strong viewpoint and illumination changes as the images can be taken at different points in time and by different cameras. In this scenario, this assumption is violated and using handcrafted rather than learned features might significantly affect performance. This limitation is currently not discussed.

W2) Some of the results of the paper seem rather predictable to me. As a result, some of the claims made in the paper seem too strong:
a) I would have been surprised if the proposed approach would not have generalized from synthetic to real data. The used features are handcrafted and will thus not be affected by the domain gap. Learning cost aggregation is a higher-level task that operates on feature similarities, i.e., an abstract representation. Learnable cost aggregation methods have been shown to generalize, e.g., [Seki & Pollefeys, SGM-Nets: Semi-global matching with neural networks, CVPR 2017] propose a trainable version of the semi-global matching framework from [1] and show that it generalizes from synthetic to real data (for both learned and handcrafted features).
b) Given the SGM-Nets paper, which uses a trainable cost aggregation strategy and has been evaluated with a non-parametric cost volume computed from handcrafted features, I think that the statement that "this paper proposes to rethink the role of DNNs. It presents a method that casts stereo matching as a cost aggregation problem (solved by training a DNN) over a non-parametric cost volume (that truly focuses on matching) with parametric contextual features" is too strong.
c) One central question the paper asks is how well learned stereo matchers actually learn to establish matches. Based on the adversarial attacks, and the limited generalization from synthetic to real data, the paper concludes that "DNNs may not learn to perform matching well in the sense that they should otherwise achieve potentially even better after stereo-constrained perturbations are introduced". Yet, I am not sure how valid this conclusion is: [Savinov et al., Matching neural paths: transfer from recognition to correspondence search, NIPS 2017] show that DNNs trained for classification can be used for matching. In particular, they show interesting qualitative results on stylized versions of the stereo input images, i.e., inputs that are heavily perturbed. Their results suggest that DNN features can be used to for robust matching.
d) The paper states that "[22] studied adversarial patch based attacks in optical flow, which is inherently different to attacking stereo matching due to the underlyingly different matching nature of the problems." I do not agree with this statement. The binocular stereo matching problem considered in this paper, where both images are rectified with respect to each other, is a special case of the optical flow problem (where the flow in the y-direction of the image is 0). As such, the relevance of the adversarial attacks from [22] should be discussed in more detail.

W3) Multiple times, it is necessary to compare results from tables on different pages, which makes is harder than necessary to follow the paper.

Overall, this is a solid paper. I will consider raising my rating if the rebuttal successfully addresses the weaknesses listed above.

**Post-rebuttal comments**

The rebuttal successfully addressed my concerns and I thus increase my rating.

---

> ### Author Response · Authors · 2022-08-02
> **Response 1/1**
>
> Thank you for your time reviewing our paper. In the following, we address it point by point.
>
> **Comments**: Connections and differences with the SGM-Net.
>
> **Response**: Thank you for pointing out this interesting work and we will discuss it in the revision. We will also revise some of the claims to be more precise. Our approach is different from this method in several principal ways: (i) Our method contains a parametric context extractor on the reference image, which largely improves the in-domain accuracy. This is important because ``matching" alone cannot resolve occluded pixels well. SGM-Net mainly shows accuracy on non-occluded regions. (ii) Our method does not constrain the form of the cost aggregation component, whereas SGM-Net enforces the specific form of SGM, similarly as the GANet we included in our comparison. We show that just by using non-parametric inputs, we can train a more generalizable cost aggregation component. We also validate this result using the hand-crafted sum of absolute differences as shown in Table 7. (iii) Though this paper shows the results of using traditional hand-crafted features as a comparison (for the synthetic data only), their best and main results were computed using CNN features as in [16]. Their Sim2Real results were using the CNN features instead of the hand-crafted ones. Therefore, it is not their intention to advocate the non-parametric cost volumes.
>
>
>
> **Comments**: The results by Savinov et al. (NeurIPS17) suggest that DNN features can be used to for robust matching.
>
> **Response**: Thank you for pointing out this interesting work and we will discuss it in revision. This argument can be best demonstrated in the toy example in Appendix C.1, which shows that state-of-the-art DNNs fail easily on adversarial patches that have constant disparities, no matter what the disparity is set to. However, they all perform well on clean random patches. This means that these DNNs work fine in most cases, but there are "loopholes" in the system. If they are doing strict matches, this case of constant disparity patch can be easily solved. On the other hand, by enforcing "matching", our method shows much stronger robustness.
>
>
> **Comments**: the relevance of the adversarial attacks from [22] should be discussed in more detail.
>
> **Response**: Thanks for the suggestion and we will provide more details in the revision. Attacks from [22] mainly focus on showing that an adversarial patch in consecutive frames, even when it is not moving, can easily attack state-of-the-art optical flow models. This finding is important, however, the authors did not propose effective ways to defend against such attacks. On the other hand, we not only show the vulnerability of stereo matching methods, but also propose a method that is robust against different kinds of adversarial attacks as shown in the experiments. However, since optical flow is inherently easier to attack than stereo matching due to the larger search space and fewer constraints (more occlusions, larger color differences between the same physical points, etc.), whether our approach can be extended to defend optical flow is another research problem to consider.
>
> **Comments**: Limitations for Multi-View stereo problems.
>
> **Response**: Thanks for the suggestion and we will discuss more regarding the limitations of extending this approach to multi-view stereo problems.

---

> > ### Comment · Reviewer_WPxi · 2022-08-07
> > **Re: Response 1/1**
> >
> > Thank you very much for the detailed answers. Please find my comments and follow-up questions below.
> >
> > > (i) Our method contains a parametric context extractor on the reference image, which largely improves the in-domain accuracy. This is important because ``matching" alone cannot resolve occluded pixels well. SGM-Net mainly shows accuracy on non-occluded regions.
> >
> > SGM finds paths through the cost volume under some (learned) smoothness prior. For occluded pixels, there won't be a clear minimum in the matching costs and SGM will thus propagate depth (relying more on the smoothness prior than the actual matching costs). As such, the semi-global nature of SGM helps it to handle occluded pixels.
> >
> > > (ii) Our method does not constrain the form of the cost aggregation component, whereas SGM-Net enforces the specific form of SGM, similarly as the GANet we included in our comparison. We show that just by using non-parametric inputs, we can train a more generalizable cost aggregation component. We also validate this result using the hand-crafted sum of absolute differences as shown in Table 7.
> >
> > I am not completely sure I understand. Is the argument here that the cost aggregation part generalizes better than previous approaches? Does this include a comparison with SGM and SGM-Net?
> >
> > > (iii) Though this paper shows the results of using traditional hand-crafted features as a comparison (for the synthetic data only), their best and main results were computed using CNN features as in [16]. Their Sim2Real results were using the CNN features instead of the hand-crafted ones. Therefore, it is not their intention to advocate the non-parametric cost volumes.
> >
> > My point was not that the SGM-Net paper advocates the use of non-parametric cost volumes, but that they do consider non-parametric in their evaluation. As such, they do model "stereo matching as a cost aggregation problem (solved by training a DNN) over a non-parametric cost volume". Claiming this as a novel view thus seems to strong to me.
> >
> >
> > > Thank you for pointing out this interesting work and we will discuss it in revision. This argument can be best demonstrated in the toy example in Appendix C.1, which shows that state-of-the-art DNNs fail easily on adversarial patches that have constant disparities, no matter what the disparity is set to. However, they all perform well on clean random patches. This means that these DNNs work fine in most cases, but there are "loopholes" in the system. If they are doing strict matches, this case of constant disparity patch can be easily solved. On the other hand, by enforcing "matching", our method shows much stronger robustness.
> >
> > I agree that existing DNNs are susceptible to adversarial attacks. Yet, the statement "DNNs may not learn to perform matching well in the sense that they should otherwise achieve potentially even better after stereo-constrained perturbations are introduced" seems to be too strong to me as there is other work (e.g., Savinov et al.) that shows that existing DNNs can be quite robust under very strong perturbations (which to me suggests that they can perform matching quite well). This was what I wanted to point out by referring to Savinov et al.
> >
> > > Thanks for the suggestion and we will discuss more regarding the limitations of extending this approach to multi-view stereo problems.
> >
> > Would it be possible to get a summary / preview of this discussion?

---

> > > ### Author Response · Authors · 2022-08-08
> > > **Re: Re: Response 1/2**
> > >
> > > Thank you very much for further elaborating on your comments and for providing more insights.  We apologize for not fully addressing some of your original questions.
> > >
> > > **Comments**:  SGM and SGM-Nets do model "stereo matching as a cost aggregation problem (solved by training a DNN) over a non-parametric cost volume". Claiming this as a novel view thus seems too strong.
> > >
> > > > SGM-Nets by Seki & Pollefeys do provide an elegant and well-designed integration between exploiting the SGM for more robust cost aggregation (handling occlusion) and leveraging DNNs for learnable hyper-parameter optimization (P1 and P2 in SGM). In terms of the integration strategy, the proposed work shares similar spirits and motivations with SGM-Nets. The main differences lie in two aspects:
> > >
> > > > (i) By design, SGM-Nets are specific to the SGM realization for cost aggregation. Despite of the highly expressive power of the SGM optimization and even though the two hyper-parameters P1 and P2 at each pixel (with respect to a limited number of directions, e.g. 4 in SGM-Nets) are learned via DNNs, the SGM realization for cost aggregation may still be the "bottleneck" in general. In contrast, in our proposed method, we leverage a stack of Hourglass building blocks (equipped with 3D convolution) as the generic fully-trainable cost aggregation component, without resorting to specific optimization structures and resulting in more "learnable ingredients" than SGM.  On the other hand, the inductive bias of SGM may not be easily learned by the generic Hourglass stack, even though we observe that it works reasonably well in our experiments.
> > >
> > > > (ii)  SGM-Nets did not explore the integration between non-parametric cost volume and parametric context features (from the reference image only and learned from scratch), which is a non-trivial component in terms of our experimental observations.
> > >
> > > > In sum, for the statement in the submission that "this paper proposes to rethink the role of DNNs. It presents a method that casts stereo matching as a cost aggregation problem (solved by training a DNN) over a non-parametric cost volume (that truly focuses on matching) with parametric contextual features",   we will revise it to be **"this paper proposes DNN-contextualized non-parametric cost-volume. On top of that, it revisits the perspective of learning the cost aggregation via DNNs for stereo matching, and presents a simple yet expressive design that is fully end-to-end trainable, without resorting to specific aggregation inductive biases."**. We will continue to polish the statement to be more precise under the context of SGM-Nets and other related works.
> > >
> > > **Comments**: I agree that existing DNNs are susceptible to adversarial attacks. Yet, the statement "DNNs may not learn to perform matching well in the sense that they should otherwise achieve potentially even better after stereo-constrained perturbations are introduced" seems to be too strong to me as there is other work (e.g., Savinov et al.) that shows that existing DNNs can be quite robust under very strong perturbations (which to me suggests that they can perform matching quite well). This was what I wanted to point out by referring to Savinov et al.
> > >
> > > > The matching neural path method by Savinoy et al leverages neural paths in DNNs trained for classification for matching, which exploit low, middle and high level features jointly and thus are robust.  The style transfer experiments conducted by Savinoy et al are impressive. We consider white-box attacks using the PGD method in the paper.  Consider that DNNs trained for classification are also significantly vulnerable to adversarial attacks, we suspect that we could still learn to "fool" those DNNs in neural-path-based matching using the white-box PGD method. In the meanwhile, for the statement in the submission that "DNNs may not learn to perform matching well in the sense that they should otherwise achieve potentially even better after stereo-constrained perturbations are introduced", we refer to state-of-the-art DNN based stereo matching methods in which DNNs are trained from scratch.  So, we will revise the statement to be **"State-of-the-art stereo matching DNNs that are trained from scratch may not actually learn to perform matching well in the sense that they should otherwise achieve potentially even better after stereo-constrained perturbations are introduced."**

---

> > > > ### Author Response · Authors · 2022-08-08
> > > > **Re: Re: Response 2/2**
> > > >
> > > > **Comments**: Would it be possible to get a summary / preview of this discussion of the applicability and limitation of the proposed method for multi-view stereo matching?
> > > >
> > > > > As pointed out in your comments:  in multi-view stereo matching, there can be strong viewpoint and illumination changes as the images can be taken at different points in time and by different cameras. In this scenario, using handcrafted rather than learned features might significantly affect performance.
> > > >
> > > > > Assume that the multi-view images are calibrated.  On the one hand, we agree that using handcrafted features only will not work well considering the large appearance and structural variations. On the other hand, the proposed contextualized non-parametric cost volume and the generic Hourglass stack based cost aggregation could be potentially useful for addressing the challenges. We will be happy to investigate this in future work.
> > > >
> > > > **Comments**: Multiple times, it is necessary to compare results from tables on different pages, which makes is harder than necessary to follow the paper.
> > > >
> > > > > We realized that we have not addressed this.  We had a version that uses bigger tables summarizing different results together, and we were worried about being too congestive. We will revise the tables to be self-contained, keeping them separated for different experiments, but replicating some parts when necessary.

---

> > > > ### Comment · Reviewer_WPxi · 2022-08-10
> > > > **This resolves my concerns**
> > > >
> > > > Thank you very much for your detailed answer. This resolves my concerns.

---

### Official Review · Reviewer_3tmL · 2022-07-13

**Rating:** 6
**Confidence:** 3
**Soundness:** 3 good
**Presentation:** 2 fair
**Contribution:** 3 good

**Summary:**

The authors propose an approach for stereo depth estimation that is robust to adversarial attacks, due to its use of traditional image features instead of CNN-based features.

**Questions:**

Your approach +ctx appears to also be doing quite well and be robust to the attacks. Can you explain that a bit more?

**Ethics Review Area:**

["I don’t know"]

**Limitations:**

The limitations of the approach are not discussed much.

**Strengths And Weaknesses:**

Positive:
- The results appear to be quite good.
- The use of non-DNN features is quite interesting, and the experimental section is reasonably good at demonstrating their value.

Negative:
- While I am not an expert in attacks of stereo networks, the way the authors study the brittleness of CNN feature backbones feels unfair — they select a neural net attack strategy and find an approach (non-CNN-based) where this attack does not work as well. They do not however say if / how the method would behave with a non-CNN-based attack strategy, or, indeed, one designed for their matcher.
- Tables are difficult to read — e.g. Tables 1 and 2 show various columns (eg EPE, Bad 1.0, etc) without explaining what these metrics represent or what units they use. These might be somewhat standard in stereo, but, nevertheless, should be explained with those tables (instead of just towards the end of the paper in section 4).
- The cost aggregation stage does not seem to be described much.
- The authors state that in certain configs their approach is “will be unattackable since the gradient flows are completely blocked”. This should be explained more, since I’d expect differentiable cost aggregation stages to still be attackable …
- The paper is a bit confused about its own objectives — is it about adversarial robustness or is it about generalizability …

---

> ### Author Response · Authors · 2022-08-02
> **Response 1/1**
>
> Thank you for your time reviewing our paper. In the following, we address it point by point.
>
> **Comments**: Your approach +ctx appears to also be doing quite well and be robust to the attacks. Can you explain that a bit more?
>
> **Response**: Thanks for pointing out the effectiveness of the context extractor. We will explain this in more details in revision. We also consider this as a surprising and important finding in stereo matching because the usage of context extractor improves accuracy while maintaining robustness. While the matching component alone can explain away most of the non-occluded texture-rich regions, training the context extractor together with the matching component can encourage the context extractor to focus on context cues for regions where matching alone suffers, such as occluded, textureless or repetitive regions. To attack the matching component and the context extractor at the same time, an attaching method (such as the PGD used in the submission) must find ways to alter both the cost volume and the context cues from the reference image. The fact that the robustness is not affected significantly shows that the alternation of the context cues either contradicts the alternation of the matching component, or being too weak compared to the matching component. In other words, the gradients from the context extractor have different directions or much smaller magnitudes compared to the gradients from the matching component. This finding shows a way to improve accuracy without sacrificing robustness in stereo matching.
>
>
> **Comments**: The authors state that in certain configs their approach is “will be unattackable since the gradient flows are completely blocked”. This should be explained more, since I’d expect differentiable cost aggregation stages to still be attackable …}
>
> **Response**: Thanks for the suggestion and we will describe in more details in the revision. To compute the gradients of the error with respect to the image pixels, we will need to back-propagate the gradients through all the layers until they reach the input image. In the setting without the context extractor, these gradients will be blocked by the non-differentiable operator, since we compare each pixel to the center within a patch (i.e. the census transform). Therefore, if we do not provide ways to approximate the gradients of ">", the gradients with respect to the image pixels will be none.
>
>
> **Comments**:  While I am not an expert in attacks of stereo networks, the way the authors study the brittleness of CNN feature backbones feels unfair...
>
> **Response**: To our knowledge, currently there are no studies on adversarial attacks for traditional stereo matching methods, such as SGM. Adversarial attacks usually refer to the issue that there exists some small perturbations of an image, or some perturbations of a small portion of the image, which can easily deceive DNNs but not humans. Therefore, it is usually not a problem for traditional 3D vision methods. The attack method PGD that we used is also a standard way to test adversarial robustness for all DNNs in a white-box attack setting.
>
>
> **Comments**: The paper is a bit confused about its own objectives — is it about adversarial robustness or is it about generalizability.
>
> **Response**: It is our main goal to tackle both problems through the same network. These two problems are both important for autonomous driving applications.

---

> > ### Comment · Reviewer_3tmL · 2022-08-10
> > **Thank you for the response**
> >
> > Thank you for your response. Based on your comments and the other reviews I am raising my score to weak accept.

---

### Meta-Review · Area_Chair_Zh9Z · 2022-08-21

**Recommendation:** Accept
**Confidence:** Certain

**Metareview:**

This paper addresses the problem of robustness in stereo-matching. It has been reviewed by several knowledgeable reviewers with extensive experience in stereo-matching and learning for stereo. The majority consensus from the reviews was that the paper will be of interest to the community and should be accepted. This meta-review agrees, and recommends acceptance.

However, as noted by the reviewers, there are some issues with the text that need to be fixed e.g.:
Lack of focus in aspects of the presentation (3tmL)
Lacking descriptions of the method and evaluation (3tmL)
Lack of discussion of the multi-view (not time synchronized) setting (WPxi)
Difficult to compare the results across Tables (WPxi)
Missing discussion of Cai et al. 3DV 2020 (RUGj)


Finally, while HFdJ was less supportive of the paper in their original review, they did not provide references to address their claims that the "The cost aggregation problem perspective has been widely exploited." Furthermore, they did not engage in the discussion to further expand on their concerns. Given this, less weight was placed on their comments.


**Award:**

No

---

### Decision · Program_Chairs · 2022-09-14

Accept